# TEXT2REWARD: REWARD SHAPING WITH LANGUAGE MODELS FOR REINFORCEMENT LEARNING

**Tianbao Xie**[*][♠]  **Siheng Zhao**[*][♠♡]  **Chen Henry Wu**[◇]  **Yitao Liu**[♠]  **Qian Luo**[♠]
**Victor Zhong**[♣♦]  **Yanchao Yang**[†♠]  **Tao Yu**[†♠]
[♠]The University of Hong Kong  [♡]Nanjing University  [◇]Carnegie Mellon University
[♣]Microsoft Research  [♦]University of Waterloo

## ABSTRACT

Designing reward functions is a longstanding challenge in reinforcement learning (RL); it requires specialized knowledge or domain data, leading to high costs for development. To address this, we introduce TEXT2REWARD, a data-free framework that automates the generation and shaping of dense reward functions based on large language models (LLMs). Given a goal described in natural language, TEXT2REWARD generates shaped dense reward functions as an executable program grounded in a compact representation of the environment. Unlike inverse RL and recent work that uses LLMs to write sparse reward codes or unshaped dense rewards with a constant function across timesteps, TEXT2REWARD produces interpretable, free-form dense reward codes that cover a wide range of tasks, utilize existing packages, and allow iterative refinement with human feedback. We evaluate TEXT2REWARD on two robotic manipulation benchmarks (MANISKILL2, METAWORLD) and two locomotion environments of MUJOCO. On 13 of the 17 manipulation tasks, policies trained with generated reward codes achieve similar or better task success rates and convergence speed than expert-written reward codes. For locomotion tasks, our method learns six novel locomotion behaviors with a success rate exceeding 94%. Furthermore, we show that the policies trained in the simulator with our method can be deployed in the real world. Finally, TEXT2REWARD further improves the policies by refining their reward functions with human feedback. Video results are available at https://text-to-reward.github.io.

## 1 INTRODUCTION

Reward shaping (Ng et al., 1999) remains a long-standing challenge in reinforcement learning (RL); it aims to design reward functions that guide an agent towards desired behaviors more efficiently. Traditionally, reward shaping is often done by manually designing rewards based on expert intuition and heuristics, while it is a time-consuming process that demands expertise and can be sub-optimal. Inverse reinforcement learning (IRL) (Ziebart et al., 2008; Wulfmeier et al., 2016; Finn et al., 2016) and preference learning (Christiano et al., 2017; Ibarz et al., 2018; Lee et al., 2021; Park et al., 2022) have emerged as potential solutions to reward shaping. A reward model is learned from human demonstrations or preference-based feedback. However, both strategies still require considerable human effort or data collection; also, the neural network-based reward models are not interpretable and cannot be generalized out of the domains of the training data.

This paper introduces a novel framework, TEXT2REWARD, to generate and shape dense reward code based on goal descriptions. Given an RL goal (e.g., "push the chair to the marked position"), TEXT2REWARD generates dense reward code (Figure 1 middle) based on large language models (LLMs), grounded on a compact, Pythonic representation of the environment (Figure 1 left). The dense reward code is then used by an RL algorithm such as PPO (Schulman et al., 2017) and SAC (Haarnoja et al., 2018) to train a policy (Figure 1 right). Different from inverse RL, TEXT2REWARD is data-free and generates symbolic reward with high interpretability. Different from recent work (Yu et al., 2023) that used LLMs to write unshaped reward code with hand-designed APIs, our free-form shaped dense reward code has a wider coverage of tasks and can utilize established

---

[*] Equal contribution. Work mainly done at the University of Hong Kong. [†]Corresponding author.

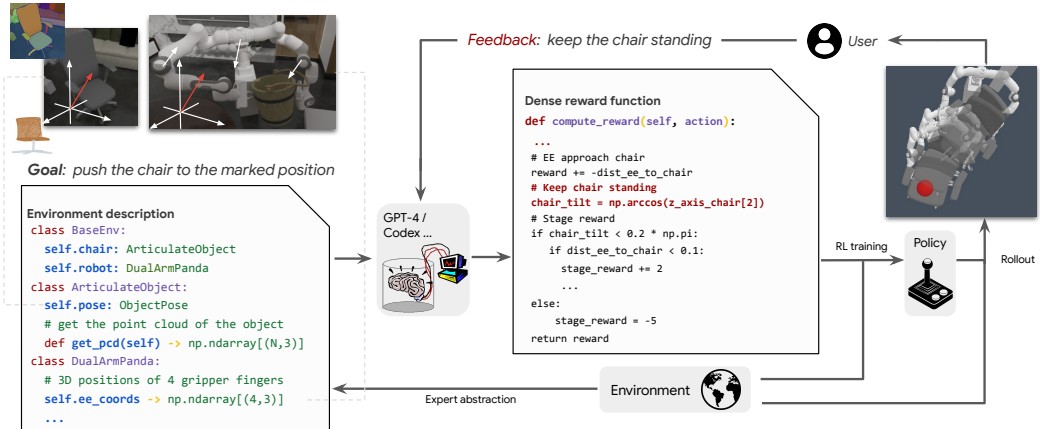

Figure 1: An overview of TEXT2REWARD of three stages: *Expert Abstraction* provides an abstraction of the environment as a hierarchy of Pythonic classes. *User Instruction* describes the goal to be achieved in natural language. *User Feedback* allows users to summarize the failure mode or their preferences, which are used to improve the reward code.

coding packages (e.g. NumPy operations over point clouds and agent positions). Finally, given the sensitivity of RL training and the ambiguity of language, the RL policy may fail to achieve the goal or achieve it in unintended ways. TEXT2REWARD addresses this problem by executing the learned policy in the environment, requesting human feedback, and refining the reward accordingly.

We conduct systematic experiments on two robotics manipulation benchmarks (MANISKILL2 (Gu et al., 2023), METAWORLD (Yu et al., 2020)) and two locomotion environments of MUJOCO (Brockman et al., 2016), as cases. On 13 out of 17 manipulation tasks, policies trained with our generated reward code achieve comparable or better success rates and convergence speed than the ground truth reward code carefully tuned by human experts. For locomotion, TEXT2REWARD learns 6 novel locomotion behaviors with over 94% success rate. We also demonstrate that the policy trained in the simulator can be deployed on a real Franka Panda robot. With human feedback of less than 3 iterations, our method can iteratively improve the success rate of learned policy from 0 to almost 100%, as well as resolve task ambiguity. In summary, the experimental results demonstrated that TEXT2REWARD can generate generalizable and interpretable dense reward code, enabling a wide coverage of RL tasks and a human-in-the-loop pipeline. We hope that the results can inspire further explorations in the intersection of reinforcement learning and code generation.

## 2 APPROACH

### 2.1 BACKGROUND

**Reward code** Reinforcement learning (RL) aims to learn a policy that maximizes the expected reward in an episode. To train a policy to achieve a goal, the key is to design a reward function that specifies the goal. The reward function can take various forms such as a neural network or a piece of reward code. In this paper, we focus on the reward code given its interpretability. In this case, the observation and the action are represented as variables, such that the reward does not need to handle perception – it only reasons about abstract variables and APIs in code.

**Reward shaping** Reinforcement learning from task completion rewards is difficult because the reward signals are sparse and delayed (Sutton & Barto, 2005). A shaped dense reward function is useful since it encourages key intermediate steps and regularization that help achieve the goal. In the form of code, the shaped dense reward can take different functional forms at each timestep, instead of being constant across timesteps or just at the end of the episode.

## 2.2 ZERO-SHOT AND FEW-SHOT DENSE REWARD GENERATION

In this part, we describe the core of TEXT2REWARD for zero-shot and few-shot dense reward generation. Detailed prompt examples can be found in the Appendix C. Interactive generation is described in the next subsection.

**Instruction**   The instruction is a natural language sentence that describes what we want the agent to achieve (e.g. "push the chair to the marked position"). It can be provided by the user, or it can be one of the subgoals for a long-horizon task, planned by the LLM.

**Environment abstraction**   To ground reward generation in an environment, it is necessary for the model to know how object states are represented in the environment, such as the configuration of robots and objects, and what functions can be called. We adopt a compact representation in Pythonic style as shown in Figure 1, which utilizes Python class, typing, and comment. Compared to listing all environment-specific information in the list or table format, Pythonic representation has a higher level of abstraction and allows us to write general, reusable prompts across different environments. Moreover, this Pythonic representation is prevalent in LLMs pre-training data, making it easier for the LLM to understand the environment.

**Background knowledge**   Generating dense reward codes can be challenging for LLMs due to the scarcity of data in these domains. Recent works have shown the benefits of providing relevant function information and usage examples to facilitate code generation (Shi et al., 2022; Zhou et al., 2022). Inspired by them, we provide functions that can be helpful in this environment as background knowledge (e.g., NumPy/SciPy functions for pairwise distance and quaternion computation, specified by their input and output types and natural language explanations).

**Few-shot examples**   Providing relevant examples as input has been shown to be useful in helping LLMs solve tasks. We assume access to a pool of pairs of instructions and verified reward codes. The library can be initialized by experts and then continually extended by our generated dense reward code. We utilize the sentence embedding model from Su et al. (2022) to encode each instruction. Given a new instruction, we use the embedding to retrieve the top-$k$ similar instructions and concatenate the instruction-code pairs as few-shot examples. We set the $k$ to 1 since context length limits and filter out the oracle code of the task from the retrieved pool to make sure that the LLMs do not cheat.

**Reducing error with code execution**   Once the reward code is generated, we execute the code in the code interpreter. This step may give us valuable feedback, e.g., syntax errors and runtime errors (e.g., shape mismatch between matrices). In line with previous works (Le et al., 2022; Olausson et al., 2023), we utilize the feedback from code execution as a tool for ongoing refinement within the LLM. This iterative process fosters the systematic rectification of errors and continues until the code is devoid of errors. Our experiments show that this step decreases error rates from 10% to near zero.

## 2.3 IMPROVING REWARD CODE FROM HUMAN FEEDBACK

Humans seldom specify precise intent in a single interaction. In an optimistic scenario, the initial generated reward functions may be semantically correct but practically sub-optimal. For instance, users instructing a robot to open a cabinet may not specify whether to pull the handle or the edge of the door. While both methods open the cabinet, the former is preferable because it is less likely to damage the furniture and the robot. In a pessimistic scenario, the initially generated reward function may be too difficult to accomplish. For instance, telling a robot to "clean up the desk" results in a more difficult learning process than telling the robot to "pick up items on the desk and then put them in the drawer below". While both descriptions specify the same intent, the latter provides intermediate objectives that simplify the learning problem.

To address the problem of under-specified instructions resulting in sub-optimal reward functions, TEXT2REWARD actively requests human feedback from users to improve the generated reward functions. After every RL training cycle, the users are provided with rollout videos of task execution by the current policy. Users then offer critical insights and feedback based on the video, identifying areas of improvement or errors. This feedback is integrated into subsequent prompts to generate more refined and efficient reward functions. In the first example of opening a cabinet, the

user may say "use the door handles" to discourage the robot from damaging itself and the furniture by opening using the door edges. In the second example of cleaning a desk, the user may say "pick up the items and store them in the drawer" to encourage the robot to solve sub-tasks. It is noteworthy that this setup encourages the participation of general users, devoid of expertise in programming or RL, enabling a democratized approach to optimizing system functionality through natural language instructions, thus eliminating the necessity for expert intervention.

## 3 EXPERIMENT SETUP

We evaluate TEXT2REWARD on manipulation and locomotion tasks across three environments: METAWORLD, MANISKILL2, and Gym MUJOCO. We use GPT-4[1] as the LLM to demonstrate our method and further includes an examination of other open-source models, as presented in Appendix F. This is done not only to ensure reproducibility but also to clarify the complexity of the task at hand. We choose the RL algorithm (PPO or SAC) and set default hyper-parameters according to the performance of human-written reward, and fix that in all experiments on this task to do RL training. Experiment hyperparameters are listed in Appendix A.

### 3.1 MANIPULATION TASKS

We demonstrate manipulation on METAWORLD, a commonly used benchmark for Multi-task Robotics Learning and Preference-based Reinforcement Learning (Nair et al., 2022; Lee et al., 2021; Hejna III & Sadigh, 2023), and MANISKILL2, a platform showcasing a diverse range of object manipulation tasks executed within environments with realistic physical simulations. We evaluate a diverse set of manipulation tasks including pick-and-place, assembly, articulated object manipulation with revolute or sliding joint, and mobile manipulation. For all tasks, we compare TEXT2REWARD with *oracle* reward functions tuned by human experts (provided in the original codebases). We also establish a baseline by adapting the prompt from Yu et al. (2023) to suit our reinforcement learning framework, as opposed to the Model Predictive Control (MPC) setting originally described in Yu et al. (2023) since designing the physics model demands a considerable amount of additional expert labor. For RL training, we tune the hyperparameters such that the oracle reward functions have the best results, and then keep them fixed when running TEXT2REWARD. The full list of tasks, corresponding input instructions, and details of simulated environments are found in Appendix B.

### 3.2 LOCOMOTION TASKS

For locomotion tasks, we demonstrate our method using Gym MUJOCO. Due to the lack of expert-written reward functions for locomotion tasks, we follow previous work (Christiano et al., 2017; Lee et al., 2021) to evaluate the policy based on human judgment of the rollout video. We develop six novel tasks in total for two different locomotion agents, Hopper (a 2D unipedal robot) and Ant (a 3D quadruped robot). The tasks include Move Forward, Front Flip and Back Flip for Hopper, as well as Move Forward, Lie Down, and Wave Leg for Ant.

### 3.3 REAL ROBOT MANIPULATION

Unlike model-based methods such as model predictive control (MPC) (Howell et al., 2022), which require further parameter adjustment, our RL agents—trained in a simulator—can be directly deployed in the real world, necessitating only minor calibration and the introduction of random noise for sim-to-real transfer. To demonstrate this benefit, as well as verify the generalization ability of RL policy trained from our generated reward, we conducted a real robot manipulation experiment with the Franka Panda robot arm. We verify our approach on two manipulation tasks: Pick Cube and Stack Cube. To obtain the object state required by our RL policy, we use the Segment Anything Model (SAM) (Kirillov et al., 2023) and a depth camera to get the estimated pose of objects. Specifically, we query SAM to segment each object in the scene. The segmentation map and the depth map together give us an incomplete point cloud. We then estimate the pose of the object based on this point cloud.

---

[1]https://platform.openai.com/docs/guides/gpt. This work mainly uses `gpt-4-0314`.

### 3.4 INTERACTIVE GENERATION WITH HUMAN FEEDBACK

We conduct human feedback on a challenging task for single-round reward code generation, Stack Cube, to investigate whether human feedback can improve or fix the reward code, enabling RL algorithms to successfully train models in a given environment. This task involves reaching the cube, grasping the cube, placing the cube on top of another cube, and releasing the cube while making it static. We sample 3 generated codes from zero-shot and few-shot methods and perform this task with two rounds of feedback. In addition, we also conduct experiments on one locomotion task Ant Lie Down, where the initial training results do not satisfy the user's preference. The general user who provides the feedback can only see the rollout video and learning curve, without any code. The authors provide feedback as per the described setup.

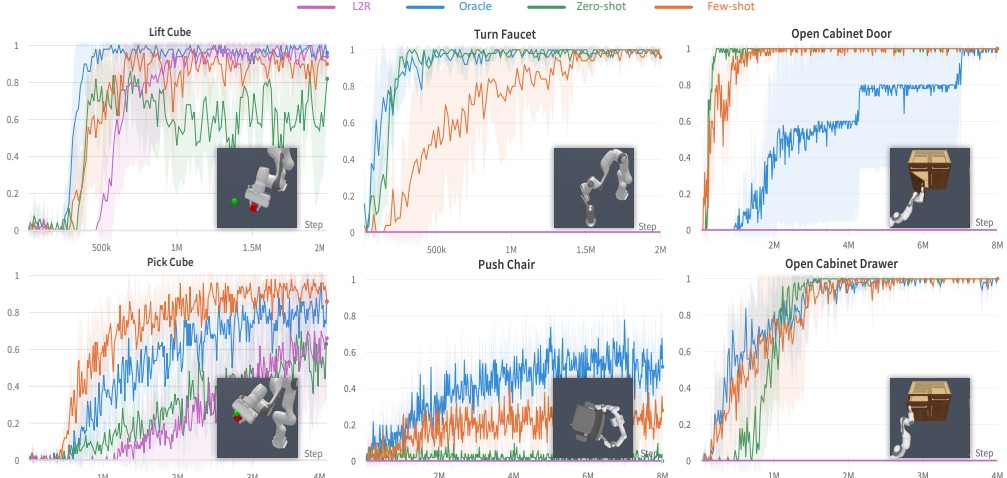

Figure 2: Learning curves on MANISKILL2 under zero-shot and few-shot reward generation settings, measured by task success rate. *L2R* means the baseline adapted from Yu et al. (2023); *Oracle* means the expert-written reward function provided by the environment; *zero-shot* and *few-shot* (*k*=1) means the reward function is generated by TEXT2REWARD w.o and w. retrieving expert-written examples from other tasks. The solid line represents the mean success rate, while the shaded regions correspond to the standard deviation, both calculated across five different random seeds.

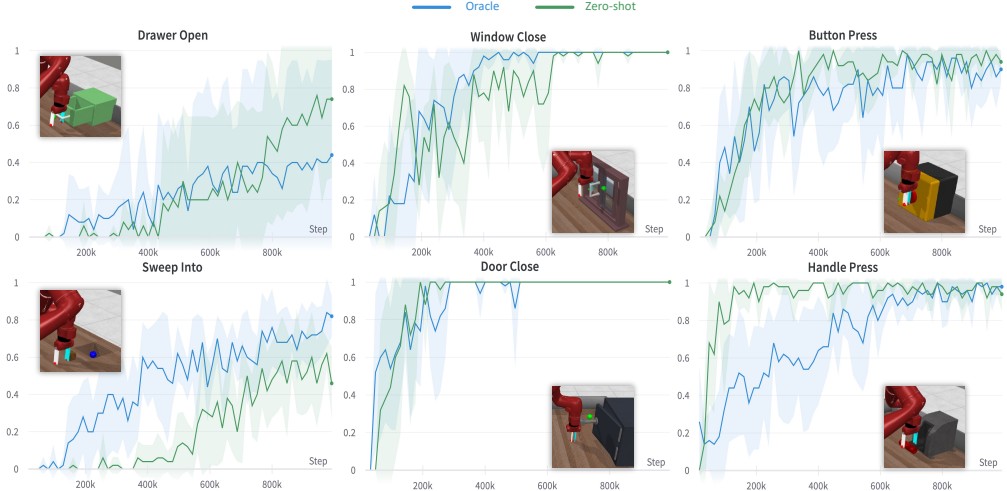

Figure 3: Learning curves on METAWORLD under zero-shot reward generation setting, measured by success rate. Following Figure 2, the solid line represents the mean success rate, while the shaded regions correspond to the standard deviation, both calculated across five different random seeds. Additional results can be found in the Appendix G.

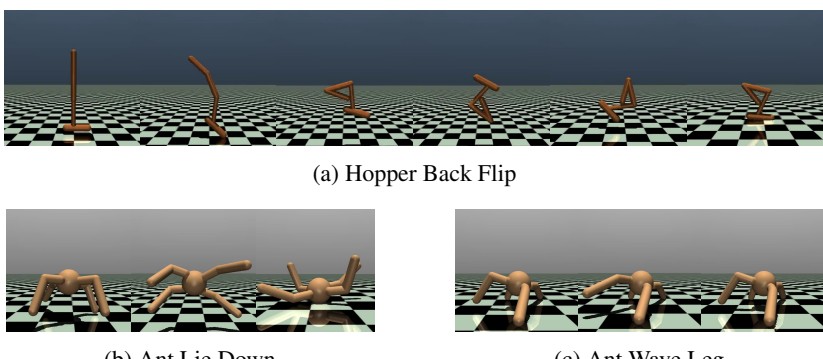

(a) Hopper Back Flip

(b) Ant Lie Down                      (c) Ant Wave Leg

Figure 4: Novel locomotion behaviors acquired through TEXT2REWARD under zero-shot reward generation setting. These images are sampled by policy rollouts in Gym MUJOCO.

## 4    RESULTS AND ANALYSIS

### 4.1    MAIN RESULTS

This section shows the results of TEXT2REWARD for robotics manipulation and locomotion. Generated reward function samples can be found in the Appendix D.

**TEXT2REWARD $\simeq$ expert-designed rewards on manipulation tasks.**    Quantitative results from the MANISKILL2 and METAWORLD environments are shown in Figures 2 and 3. In the figures, *L2R* stands for reward function generated by the baseline prompt adapted from Yu et al. (2023); *Oracle* means the expert-written dense reward function provided by the environment; *zero-shot* and *few-shot* stands for the dense reward function generated by TEXT2REWARD without human feedback under zero-shot and few-shot prompting paradigms, respectively. On 13 of the 17 tasks, the final performance (i.e., success rate after convergence and convergence speed) of TEXT2REWARD achieves comparable results to the human oracle. Surprisingly, on 4 of the 17 tasks, zero-shot and few-shot TEXT2REWARD can even outperform human oracle, in terms of either the convergence speed (e.g., Open Cabinet Door in MANISKILL2, Handle Press in METAWORLD) or the success rate (e.g., Pick Cube in MANISKILL2, Drawer Open in METAWORLD). It suggests that LLMs have the potential to draft high-quality shaped dense reward functions without any human intervention.

As demonstrated in Figure 2, the *L2R* baseline can only be effectively applied to two tasks of MANISKILL2, where it attains results comparable to those of the *zero-shot* setting. Nevertheless, *L2R* struggles with tasks involving objects that have complex surfaces and cannot be adequately described by a singular point, such as an ergonomic chair. In these tasks, the environment utilizes point cloud to denote the object's surface, a representation that *L2R* fails to model. To more comprehensively evaluate the necessity of shaped and staged dense rewards, we introduce an additional baseline in Appendix E that circumvents the aforementioned limitation of the *L2R* formulation.

Furthermore, as illustrated in Figure 2, in 2 of the 6 tasks that are not fully solvable, the few-shot paradigm markedly outperforms the zero-shot approach. This underscores the benefits of utilizing few-shot examples from our skills library in enhancing the efficacy of RL training reward functions.

**TEXT2REWARD can learn novel locomotion behaviors.**    Table 1 shows the success rate of all six tasks trained with the reward generated under the zero-shot setting, evaluated by humans watching the rollout videos. The results suggest that our method can generate dense reward functions that generalize to novel locomotion tasks. Image samples from the Gym MUJOCO environment of three selected tasks are shown in Figure 4. Corresponding full video results are available here.

**Demonstrating TEXT2REWARD on a real robot.**    Figure 5 shows the key frames of real robot manipulation on two tasks: Pick Cube and Stack Cube. Here, we use the same 7 DoF Franka Panda robot arm as MANISKILL2 simulation environment. Results suggest that the RL policy trained in the

Table 1: Success rate of locomotion tasks in Gym MUJOCO trained on reward functions generated in zero-shot setting. Each task is tested on 100 rollouts, and task success is determined by the authors who reach an agreement after reviewing the rollout videos. Generated codes are in Appendix D.2.

| Hopper | | Ant | |
|---|---|---|---|
| Task | Success Rate | Task | Success Rate |
| Move Forward | 100% | Move Forward | 94% |
| Front Flip | 99% | Lie Down | 98% |
| Back Flip | 100% | Wave Leg | 95% |

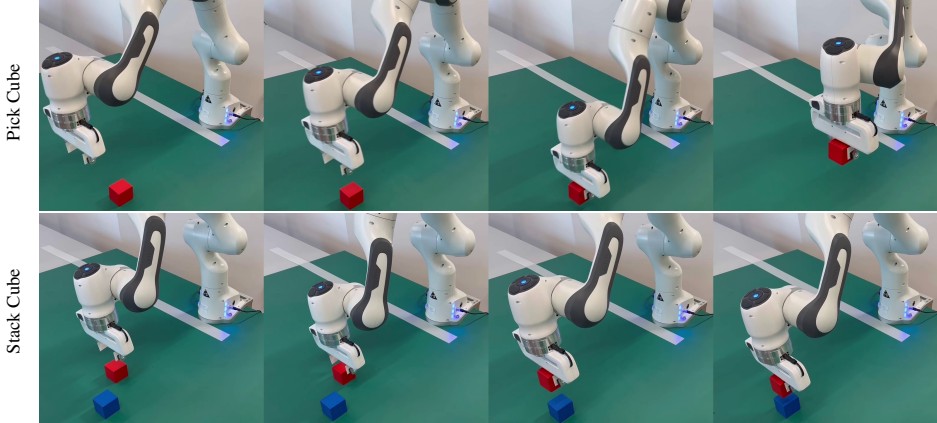

Figure 5: Sampled images for real robot manipulation on Pick Cube (i.e., pick a cube and move it to a predefined position) and Stack Cube (i.e., stack a cube onto another one).

simulator using dense reward function generated from TEXT2REWARD can be successfully deployed to the real world. Full videos of robot execution are on our project page.

**TEXT2REWARD can resolve ambiguity from human feedback.** To demonstrate the ability of TEXT2REWARD to address this problem, we show one case in which "control the Ant to lie down" itself has ambiguity in terms of the orientation of the Ant, as shown in Figure 6. After observing the training result of this instruction, the user can give the feedback in natural language, e.g., "the Ant's torso should be top down, not bottom up". Then TEXT2REWARD will regenerate the reward code and train a new policy, which successfully caters to the user's intent.

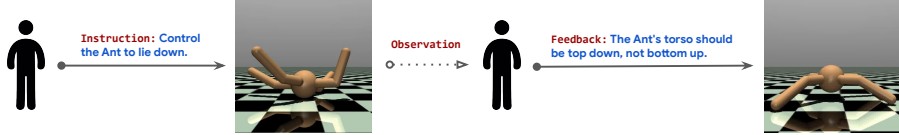

Figure 6: Interactive reward generation from human feedback. The original instruction is *control the Ant to lie down*, which is ambiguous in the orientation of the Ant. Our interactive framework allows the user to provide feedback based on the rollout observation.

**TEXT2REWARD can improve RL training from human feedback.** Given the sensitivity of RL training, sometimes single-turn generation can not generate good enough reward functions to finish the task. In these cases, TEXT2REWARD asks for human feedback on the failure mode and tries to improve the dense reward. In Figure 7, we demonstrate this on the Stack Cube task, where zero-shot and few-shot generation in a single turn fails to solve the task stably. For few-shot generation, we observed that interactive code generation with human feedback can improve the success rate from zero to one, as well as speed up the convergence speed of training. However, this improvement is prone to the quality of the reward function generated in the beginning (i.e. *iter0*). For relatively

low-quality reward functions (e.g. zero-shot generated codes), the improvement in success rate after iterations of feedback is not as pronounced as for few-shot generated codes. This problem may be solved in a sparse-to-dense reward function generation manner, which generates the stage reward first and then generates reward terms interactively. We leave this paradigm for possible future work.

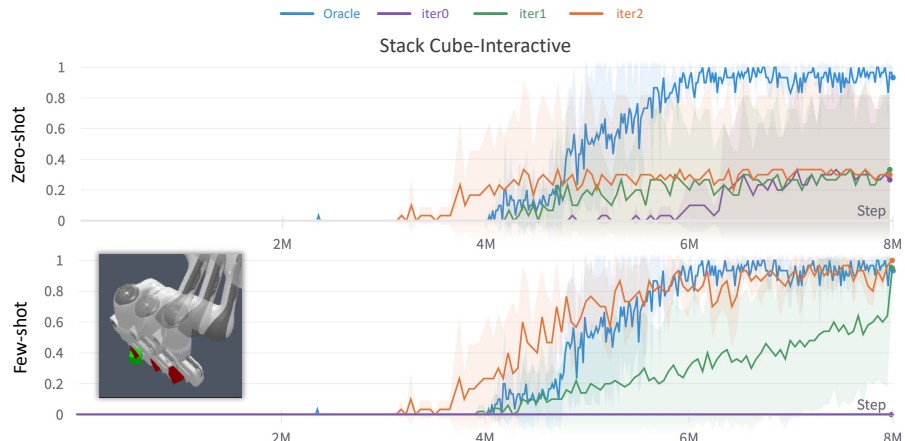

Figure 7: Training iteration vs. success rate on Stack Cube with interactive generation. *Oracle* is the reward code manually tuned by experts; *iter0* is generated by TEXT2REWARD without feedback; *iter1* and *iter2* are generated after 1 or 2 feedback iterations; *zero-shot* and *few-shot* (*k*=1) stands for how the *iter0* code is generated. The solid lines represent the mean success rate, and the shaded regions correspond to the standard deviation, with three samples.

## 4.2 QUALITATIVE ANALYSIS

In this section, we summarize the differences between the generated reward functions (zero-shot, few-shot) and the oracle reward functions. Due to space limitation, the reward functions we will refer to are in Appendix D.

**Few-shot outperforming zero-shot**  Few-shot settings generally surpass zero-shot in terms of downstream performance. In the Lift Cube and Pick Cube tasks, few-shot generated code delineates stages – approach, grasp, and lift – more clearly using conditional statements. In contrast, zero-shot code stacks and sums different stage rewards in a linear manner, reducing effectiveness. This is also demonstrated by the Push Chair task, which requires nuanced commonsense steps that zero-shot code only partially captures. Since the stage reward format is shown in the few-shot examples, GPT-4 can generalize the pattern to new tasks. Currently, this ability cannot be fully unfolded with zero-shot even if we instruct it to do so, which may be improved by the next version of LLMs.

**Zero-shot sometimes outperforming few-shot**  Our experiments also show that zero-shot learning occasionally outperforms few-shot learning, as seen in tasks like Turn Faucet and Open Doors. By checking the generated code, we found that the quality and relevance of the few-shot examples are the key. For example, in the Open Door task, few-shot learning lags as it omits a key reward term – the door's positional change – thus hindering learning. Similarly, in the Turn Faucet task, zero-shot rewards facilitate easier success by allowing one-sided gripper actions, which leads to effectiveness.

**Few-shot sometimes outperforming Oracle**  There are scenarios where few-shot learning even outperforms the oracle reward function crafted by experts. An analysis of the Open Door task reward function shows that few-shot generated code omits the final stage of stabilizing the door, simplifying the policy learning process improving the learning curve, and finishing tasks in a slightly different manner. In the Pick Cube task, few-shot and oracle codes are structurally similar, but the weight terms are different. This shows that LLMs cannot only improve reward structure and logic but also tune the hyperparameters. Although this is not our focus here, it can be a promising direction.

## 5 RELATED WORK

**Reward Shaping**   Reward shaping remains a persistent challenge in the domain of reinforcement learning (RL). Traditionally, handcrafted reward functions are employed, yet crafting precise reward functions is a time-consuming process that demands expertise. Inverse reinforcement learning (IRL) emerges as a potential solution, where a non-linear reward model is recovered from expert trajectories to facilitate RL learning (Ziebart et al., 2008; Wulfmeier et al., 2016; Finn et al., 2016). However, this technique necessitates a large amount of high-quality trajectory data, which can be elusive for complex and rare tasks. An alternative approach is preference learning, which develops a reward model based on human preferences (Christiano et al., 2017; Ibarz et al., 2018; Lee et al., 2021; Park et al., 2022; Zhu et al., 2023). In this method, humans distinguish preferences between pairs of actions, upon which a reward model is constructed utilizing the preference data. Nonetheless, this strategy still requires some human-annotated preference data which is expensive or even hard to collect in some cases. Both of these prevalent approaches to reward shaping demand extensive high-quality data, resulting in compromised generalizability and low efficiency. In contrast, TEXT2REWARD excels with limited (or even zero) data input and can be easily generalized to new tasks in the environment.

**Language Models in Reinforcement Learning**   Large Language Models (LLMs) have exhibited remarkable reasoning and planning capabilities (Wei et al., 2022; Huang et al., 2022a). Recent works have shown that the knowledge in LLMs can be helpful for RL and can transform the data-driven policy network acquisition paradigm (Carta et al., 2023; Wu et al., 2023). This trend sees LLM-powered autonomous agents coming to the fore, with a growing body of approaches that use LLMs as policy networks during the RL process, indicating a promising trajectory in this field (Yao et al., 2022; Shinn et al., 2023; Lin et al., 2023; Wang et al., 2023; Xu et al., 2023; Yao et al., 2023; Hao et al., 2023). Instead of directly using LLMs as the policy model or the reward model, TEXT2REWARD generates shaped dense reward code to train RL policies, which has an advantage in terms of the flexibility of agent model type and inference efficiency.

**Language Models for Robotics**   Utilizing LLMs for embodied applications emerges as a popular trend of research, and typical directions include planning and reasoning through language model generation (Ahn et al., 2022; Zeng et al., 2022; Liang et al., 2022; Huang et al., 2022b; Singh et al., 2023; Song et al., 2022). Recent works have harnessed the capabilities of LLMs to assist in the learning of primitive tasks (Brohan et al., 2022; Huang et al., 2023; Brohan et al., 2023; Mu et al., 2023), by finetuning LLMs on robotic trajectories to predict primitive actions while TEXT2REWARD generates reward codes to learn smaller policy networks. A recent work, L2R (Yu et al., 2023), combines reward generation and Model Predictive Control (MPC) to synthesize robotic actions. Although such rewards work well for many tasks with a well-designed MPC, our experiments show that RL training is challenging for unshaped rewards, especially on complex tasks. Different from them, TEXT2REWARD adopts more flexible program structures such as *if-else* conditions and point cloud queries to offer higher flexibility and work for tasks that unshaped dense rewards tend to fail.

## 6 CONCLUSION

We proposed TEXT2REWARD, an interactive reward code generation framework that uses LLMs to automate reward shaping for reinforcement learning. Our experiments showcased the effectiveness of our approach, as the RL policies trained with our generated reward codes were able to match or even surpass the performance of those trained with expert-designed codes in the majority of tasks. We also showcased real-world applicability by deploying a policy trained in a simulator on a real robot. By incorporating human feedback, our approach iteratively refines the generated reward codes, addressing the challenge of language ambiguity and improving the success rates of learned policies. This interactive learning process allows for better alignment with human needs and preferences, leading to more robust and efficient reinforcement learning solutions.

In conclusion, TEXT2REWARD demonstrates the effectiveness of using natural language to transform human intentions and LLMs knowledge into reward functions, then policy functions. We hope that our work may serve as an inspiration for researchers across various disciplines, including but not limited to reinforcement learning and code generation, to further investigate this promising intersection of fields and contribute to the ongoing advancement of research in these areas.

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

# Appendices

## A  HYPER-PARAMETER DETAILS

In this section, we provide the hyper-parameter details used for our reward function generation and reinforcement learning backbones. For reward function generation, we base on GPT-4 [2] In the experiments of the main body, the temperature of sampling is set to 0.7 for each experiment. For reinforcement learning training, we use open-source implementations of SAC and PPO[3] algorithm, and list the hyper-parameters in Table 2 and Table 3 respectively. Their respective training environments are indicated in parentheses.

Table 2: Hyper-parameter of SAC algorithm applied to each task.

| Hyper-parameter | Value |
| --- | --- |
| Discount factor $\gamma$ | 0.99 (MetaWorld), 0.95 (ManiSkill2) |
| Target update frequency | 2 (MetaWorld), 1 (ManiSkill2) |
| Learning rate | $3e^{-4}$ |
| Train frequency | 1 (MetaWorld), 8 (ManiSkill2) |
| Soft update $\tau$ | $5e^{-3}$ |
| Gradient steps | 1 (MetaWorld), 4 (ManiSkill2) |
| Learning starts | 4000 |
| Hidden units per layer | 256 |
| Batch Size | 512 (MetaWorld), 1024 (ManiSkill2) |
| # of layers | 3 (MetaWorld), 2 (ManiSkill2) |
| Initial temperature | 0.1 (MetaWorld), 0.2 (ManiSkill2) |
| Rollout steps per episode | 500 (MetaWorld), 200 (Maniskill2) |

Table 3: Hyper-parameter of PPO algorithm applied to each task.

| Hyper-parameter | Value |
| --- | --- |
| Discount factor $\gamma$ | 0.85 (ManiSkill2), 0.99 (MuJoco) |
| # of epochs per update | 15 (ManiSkill2), 10 (MuJoco) |
| Learning rate | $3e^{-4}$ |
| # of environments | 8 |
| Batch size | 400 (ManiSkill2), 64 (MuJoco) |
| Hidden units per layer | 256 |
| Target KL divergence | 0.05 (ManiSkill2), None (MuJoco) |
| # of layers | 2 |
| # of steps per update | 3200 (ManiSkill2), 2048 (MuJoco) |
| Rollout steps per episode | 100 (ManiSkill2), 200 (MuJoco) |

We utilize multiple `g5.4xlarge` instances (1 NVIDIA A10G, 16 vCPUs, and 64 GiB memory per instance) from AWS for RL training. The time required for training a policy is approximately 0.5 hours per task for MetaWorld, and between 0.5 to 10 hours for ManiSkill2, varying with the task's difficulty. It's worth noting that we use the default environments from MetaWorld and ManiSkill2 without any speed optimization. Further enhancements such as parallel computing can be made to improve the training speed for validating reward functions generated. An early evaluation of the code semantics before starting RL training or modifying the reward during each RL training could be a promising direction to reduce costs.

## B  TASK DETAILS

In this section, we provide a full list of tasks within each simulation environment, accompanied by their corresponding language instructions. Across all tasks, we follow the default settings of their

---

[2] https://platform.openai.com/docs/guides/gpt. This work uses `gpt-4-0314`.
[3] `https://github.com/DLR-RM/stable-baselines3`

respective environments. Here, we will also briefly describe the observation space and action space characterizing these environments. For a thorough and detailed understanding, we encourage the readers to refer to the official manual associated with each environment.[4]

**MetaWorld**   In METAWORLD environment, we use a 7 DoF Sawyer robot arm with a fixed base to complete tabletop tasks. For all tasks, the observation space is a combination of the 3D position of the robot end-effector, a normalized measurement of gripper openness, the 3D position of the manipulated object, the quaternion of the object, all of the previous measurements, and the goal position. The environment adopts end-effector delta position control, which means the action space consists of the change of the end effector's 3D position, as well as the normalized torque the gripper should apply. For all tasks, the initial and target positions of the manipulated object and the initial joint positions of the robot arm are variable. The full tasks list and their corresponding instructions are in Table 4.

**ManiSkill2**   MANISKILL2 environment uses a 7 DoF Franka Panda as the default robot arm. For manipulation tasks (Lift Cube, Pick Cube, Turn Faucet and Stack Cube), we use a robot arm with a fixed base. For mobile manipulation tasks (Open Cabinet Door and Open Cabinet Drawer), we use a single robot arm with a Sciurus17 mobile base. For the mobile manipulation task Push Chair, we use a dual-arm robot arm with a Sciurus17 mobile base. For all tasks, the observation space consists of robot proprioception information (e.g. current joint positions, current joint velocities, robot base position and quaternion in the world frame) and task-specific information (e.g. goal position, end-effector position). We use end-effector delta pose control mode for this environment, which controls the change of 3D position and rotation (represented as an axis-angle in the end-effector frame). For all tasks, the initial and target positions of the manipulated object, the initial joint positions of the robot arm and physical parameters (e.g. friction and damping coefficient) are variable. The full tasks list and their corresponding instructions can be found in Table 5.

**Gym MuJoCo**   Gym MUJOCO offers a series of simulation environments that contain Ant, Half Cheetah and so, powered by MUJOCO physic engine and pre-defined XML files, each provided with a written reward for a simple task. To evaluate the ability of TEXT2REWARD to learn novel locomotion skills, we chose to use two robotic agents: Ant (a 3D quadruped robot) and Hopper (a 2D unipedal robot). Their actions are represented by the torques applied at the hinge joints, while their observations consist of positional values and velocities of different body parts. We provide the tasks list and their corresponding instructions in Table 6.

**Real Robot**   For the tasks Pick Cube and Stack Cube, we take the joint values of the robot arm, the pose of the end-effector and cubes as input. For real-world robot control, we use end-effector delta position control mode, which first predicts a delta 3D position for the gripper to move to, then utilizes inverse kinematics to iteratively solve the target joint value. To obtain the object state required by our RL policy, we used the Segment Anything Model (SAM) (Kirillov et al., 2023) and a depth camera to get the estimated pose of objects. Specifically, we query SAM to segment each object in the scene. The segmentation map and the depth map together give us an incomplete point cloud. We then estimate the pose of the object based on this point cloud.

## C   PROMPT DETAILS

**Zero-shot prompt**   To ground LLMs into robotics simulation environments, we propose a novel Pythonic prompt, which can be abstracted by the *experts* who developed the environment. This class-like prompt is a more compact representation than simply listing all environment attributes linearly, which can delete redundant information and save more tokens, and this Pythonic prompt can also better bootstrap Python reward code generation. More specifically, our prompt leverages *python class*, *class attribute*, *python typing* and *comments* to recursively define the environment. Here we provide an example of the zero-shot prompt for MANISKILL2 manipulation tasks:

```
You are an expert in robotics, reinforcement learning and code generation. We are going to use
a Franka Panda robot to complete given tasks. The action space of the robot is a normalized
```

---

[4]Official document of MANISKILL2 is at `https://haosulab.github.io/ManiSkill2/`, and official document of Gym MUJOCO is at `https://www.gymlibrary.dev/environments/mujoco/`

Table 4: Task list of METAWORLD.

| Task | Instruction |
|------|-------------|
| Drawer Open | Open a drawer by its handle. |
| Drawer Close | Close a drawer by its handle. |
| Window Open | Push and open a sliding window by its handle. |
| Window Close | Push and close a sliding window by its handle. |
| Button Press | Press a button in y coordination. |
| Sweep Into | Sweep a puck from the initial position into a hole. |
| Door Unlock | Unlock the door by rotating the lock counter-clockwise. |
| Door Close | Close a door with a revolving joint by pushing the door's handle. |
| Handle Press | Press a handle down. |
| Handle PressSide | Press a handle down sideways. |

Table 5: Task list of MANISKILL2.

| Task | Instruction |
|------|-------------|
| Lift Cube | Pick up cube A and lift it up by 0.2 meters. |
| Pick Cube | Pick up cube A and move it to the 3D goal position. |
| Turn Faucet | Turn on a faucet by rotating its handle. The task is finished when qpos of faucet handle is larger than target qpos. |
| Open Cabinet Door | A single-arm mobile robot needs to open a cabinet door. The task is finished when qpos of cabinet door is larger than target qpos. |
| Open Cabinet Drawer | A single-arm mobile robot needs to open a cabinet drawer. The task is finished when qpos of cabinet drawer is larger than target qpos. |
| Push Chair | A dual-arm mobile robot needs to push a swivel chair to a target location on the ground and prevent it from falling over. |
| Stack Cube | Pick up cube A and place it on cube B. The task is finished when cube A is on top of cube B stably (i.e. cube A is static) and isn't grasped by the gripper. |

```
`Box(-1, 1, (7,), float32)`. Now I want you to help me write a reward function for reinforcement
learning. I'll give you the attributes of the environment. You can use these class attributes
to write the reward function.

Typically, the reward function of a manipulation task is consisted of these following parts:
1. the distance between robot's gripper and our target object
2. difference between current state of object and its goal state
3. regularization of the robot's action
4. [optional] extra constraint of the target object, which is often implied by task instruction
5. [optional] extra constraint of the robot, which is often implied by task instruction
...

class BaseEnv(gym.Env):
    self.cubeA : RigidObject # cube A in the environment
    self.cubeB : RigidObject # cube B in the environment
    self.cube_half_size = 0.02 # in meters
    self.robot : PandaRobot # a Franka Panda robot

class PandaRobot:
    self.ee_pose : ObjectPose # 3D position and quaternion of robot's end-effector
    self.lfinger : LinkObject # left finger of robot's gripper
    self.rfinger : LinkObject # right finger of robot's gripper
    self.qpos : np.ndarray[(7,)] # joint position of the robot
    self.qvel : np.ndarray[(7,)] # joint velocity of the robot
    self.gripper_openness : float # openness of robot gripper, normalized range in [0, 1]
    def check_grasp(self, obj : Union[RigidObject, LinkObject], max_angle=85) -> bool
        # indicate whether robot gripper successfully grasp an object

class ObjectPose:
    self.p : np.ndarray[(3,)] # 3D position of the rigid object
    self.q : np.ndarray[(4,)] # quaternion of the rigid object
    def inv(self,) -> ObjectPose # return a `ObjectPose` class instance, which is the inverse
                                 # of the original pose
    def to_transformation_matrix(self,) -> np.ndarray[(4,4)]
        # return a [4, 4] numpy array, which is the transform matrix

class RigidObject:
    self.pose : ObjectPose # 3D position and quaternion of the rigid object
```

Table 6: Task list of Gym MUJOCO.

| Task | Instruction |
|---|---|
| Hopper Move Forward | Control the Hopper to move in the forward direction. |
| Hopper Front Flip | Control the Hopper to front flip in the forward direction. |
| Hopper Back Flip | Control the Hopper to back flip. |
| Ant Move Forward | Control the Ant to move in the forward direction. |
| Ant Wave Leg | Control the Ant to stay in place while waving its right front leg. |
| Ant Lie Down | Control the Ant to lie down. |

```
    self.velocity : np.ndarray[(3,)] # linear velocity of the rigid object
    self.angular_velocity : np.ndarray[(3,)] # angular velocity of the rigid object
    def check_static(self,) -> bool # indicate whether this rigid object is static or not

class LinkObject:
    self.pose : ObjectPose # 3D position and quaternion of the link object
    self.velocity : np.ndarray[(3,)] # linear velocity of the link object
    self.angular_velocity : np.ndarray[(3,)] # angular velocity of the link object
    self.qpos : float # position of the link object joint
    self.qvel : float # velocity of the link object joint
    self.target_qpos:float # target position of the link object joint
    def get_local_pcd(self,) -> np.ndarray[(M,3)] # get the point cloud of the link object surface
                                                   # in the local frame
    def get_world_pcd(self,) -> np.ndarray[(M,3)] # get the point cloud of the link object surface
                                                   # in the world frame

class ArticulateObject:
    self.pose : ObjectPose # 3D position and quaternion of the articulated object
    self.velocity : np.ndarray[(3,)] # linear velocity of the articulated object
    self.angular_velocity : np.ndarray[(3,)] # angular velocity of the articulated object
    self.qpos : np.ndarray[(K,)] # position of the articulated object joint
    self.qvel : np.ndarray[(K,)] # velocity of the articulated object joint
    def get_pcd(self,) -> np.ndarray[(M,3)] # point cloud of the articulated object surface
                                             # in the world frame

Additional knowledge:
1. A staged reward could make the training more stable, you can write them in a nested
   if-else statement.
2. `ObjectPose` class support multiply operator `*`, for example: `ee_pose_wrt_cubeA =
   self.cubeA.pose.inv() * self.robot.ee_pose`.
3. You can use `transforms3d.quaternions` package to do quaternion calculation, for example:
   `qinverse(quat: np.ndarray[(4,)])` for inverse of quaternion, `qmult(quat1: np.ndarray[(4,)],
   quat2: np.ndarray[(4,)])` for multiply of quaternion, `quat2axangle(quat: np.ndarray[(4,)])`
   for quaternion to angle.

I want you to fulfill the following task: {instruction}
1. please think step by step and tell me what does this task mean;
2. then write a function that formats as `def compute_dense_reward(self, action) -> float` and
   returns the `reward : float` only.
3. When write code, you can also add some comments as your thoughts.
```

**Few-shot example prompt** In order to retrieve previously acquired skills and generate better reward codes, it is necessary to provide LLMs with few-shot examples. These examples take the format of *instruction* and *reward code* pairs, which are then filled into the prompt template. Here is the few-shot example prompt template used for MANISKILL2 tasks:

```
An example:
Tasks to be fulfilled: {instruction}
Corresponding reward function:
```python
{reward_code}
```
```

**Interactive feedback prompt** Due to the inherent information bottleneck of natural language, the users' language instructions may be ambiguous, and their preferences may not be conveyed. On the other hand, single-turn reward code generation may not work for some complex tasks. So, in these cases, multi-turn generation with human feedback can be beneficial. One possible prompt is a combination of the previous round *generated code*, a *description* of the training results (learning curve, rollout videos, etc.) and the *feedback* for improvement. This is the prompt template:

```
Generated code shown as below:
```python
```

```
{generated_code}
```

Feed this reward code into the environment, and use the RL algorithm to train the policy.
After training, I can see from the robot that:
{description}

To make the code more accurate and train better robot, the feedback for improvement is:
{feedback}

Re-imagine which steps are missed or wrong.
Show me the improved code as below:
```

# D    SAMPLES OF GENERATED REWARD FUNCTION

## D.1    ZERO/FEW-SHOT/ORACLE MANIPULATION RESULTS

In this section, to help better understand the generation ability and results of TEXT2REWARD and the
difference between generated and oracle reward functions, we provide zero-shot generated, few-shot
generated and expert-written (provided by the simulation environment) reward functions of different
tasks on MANISKILL2.

Sample of *zero-shot* generated reward function on MANISKILL2 Lift Cube task:

```python
import numpy as np
from scipy.spatial import distance

def compute_dense_reward(self, action) -> float:
    # Define reward weights
    weight_dist = 0.4
    weight_lift = 0.4
    weight_grasp = 0.2

    # Initialize reward
    reward = 0.0

    # Stage 1: Approach the cube
    ee_pos = self.robot.ee_pose.p
    cube_pos = self.cubeA.pose.p
    dist_to_cube = np.linalg.norm(ee_pos - cube_pos)
    reward_dist = -weight_dist * dist_to_cube

    # Stage 2: Grasp the cube
    grasp_success = self.robot.check_grasp(self.cubeA, max_angle=30)
    reward_grasp = weight_grasp * grasp_success

    # Stage 3: Lift the cube
    lift_amount = cube_pos[2] - self.goal_height
    reward_lift = -weight_lift * np.abs(lift_amount)

    # Total reward
    reward = reward_dist + reward_grasp + reward_lift

    # Stage 4: Maintain the cube at the goal height
    if self.cubeA.pose.p[2] >= self.goal_height:
        reward += 0.1 * (self.cubeA.pose.p[2] - self.goal_height)

    # Regularize the robot's action
    # We don't want robot to take very big action, so we add a negative reward here
    reward -= 0.01 * np.linalg.norm(action)

    return reward
```

Sample of *few-shot* generated reward function on MANISKILL2 Lift Cube task:

```python
import numpy as np

def compute_dense_reward(self, action):
    reward = 0.0

    # check if cubeA is lifted 0.2 meter
    is_obj_lifted = self.cubeA.pose.p[2] >= self.cube_half_size + self.goal_height
    # check if the robot is static
    is_robot_static = np.max(np.abs(self.robot.qvel)) <= 0.2

    # if both conditions are met, the task is successful
```

```
    success = is_obj_lifted and is_robot_static

    if success:
        reward += 5
        return reward

    # calculate the distance between robot's end-effector and cubeA
    tcp_to_obj_pos = self.cubeA.pose.p - self.robot.ee_pose.p
    tcp_to_obj_dist = np.linalg.norm(tcp_to_obj_pos)
    # calculate the reaching reward, which encourages the robot to approach cubeA
    reaching_reward = 1 - np.tanh(5 * tcp_to_obj_dist)
    reward += reaching_reward

    # check if the robot has successfully grasped cubeA
    is_grasped = self.robot.check_grasp(self.cubeA, max_angle=30)
    # if grasped, add reward
    reward += 1 if is_grasped else 0.0

    if is_grasped:
        # calculate the distance between cubeA and the target height
        obj_to_goal_dist = np.abs(self.goal_height - (self.cubeA.pose.p[2] - self.cube_half_size))
        # calculate the lifting reward, which encourages the robot to lift cubeA up
        lift_reward = 1 - np.tanh(5 * obj_to_goal_dist)
        reward += lift_reward

    return reward
```

*Oracle* (expert-written) reward function on MANISKILL2 Lift Cube task:

```
def compute_dense_reward(self, action):
    reward = 0.0

    is_obj_placed = self.cubeA.pose.p[2] >= self.goal_height + self.cube_half_size
    is_robot_static = np.max(np.abs(self.robot.qvel)) <= 0.2

    success = is_obj_placed and is_robot_static

    if success:
        reward += 2.25
        return reward

    # reaching reward
    gripper_pos = self.robot.ee_pose.p
    obj_pos = self.cubeA.pose.p
    dist = np.linalg.norm(gripper_pos - obj_pos)
    reaching_reward = 1 - np.tanh(5 * dist)
    reward += reaching_reward

    is_grasped = self.robot.check_grasp(self.cubeA, max_angle=30)

    # grasp reward
    if is_grasped:
        reward += 0.25

    # lifting reward
    if is_grasped:
        lifting_reward = self.cubeA.pose.p[2] - self.cube_half_size
        lifting_reward = min(lifting_reward / self.goal_height, 1.0)
        reward += lifting_reward

    return reward
```

Sample of *zero-shot* generated reward function on MANISKILL2 Pick Cube task:

```
import numpy as np

def compute_dense_reward(self, action):
    # Normalize action
    action = np.clip(action, -1, 1)

    # Calculate distance between gripper and cube
    gripper_pos = self.robot.ee_pose.p
    cube_pos = self.cubeA.pose.p
    dist_gripper_cube = np.linalg.norm(gripper_pos - cube_pos)

    # Calculate distance between cube and goal
    goal_pos = self.goal_position
    dist_cube_goal = np.linalg.norm(goal_pos - cube_pos)

    # Check if the robot is grasping the cube
```

```
    grasping_cube = self.robot.check_grasp(self.cubeA)

    # Define reward components
    reward_dist_gripper_cube = -1.0 * dist_gripper_cube
    reward_dist_cube_goal = -1.0 * dist_cube_goal
    reward_grasping_cube = 1.0 if grasping_cube else -1.0

    # Define weights for reward components
    weight_dist_gripper_cube = 0.3
    weight_dist_cube_goal = 0.5
    weight_grasping_cube = 0.2

    # Calculate total reward
    reward = weight_dist_gripper_cube * reward_dist_gripper_cube \
            + weight_dist_cube_goal * reward_dist_cube_goal \
            + weight_grasping_cube * reward_grasping_cube

    # Regularization on action
    reward -= 0.01 * (action ** 2).sum()

    return reward
```

Sample of *few-shot* generated reward function on MANISKILL2 Pick Cube task:

```
import numpy as np

def compute_dense_reward(self, action):
    reward = 0.0

    # Check if the cube is placed at the goal position and the robot is static
    cube_at_goal = np.linalg.norm(self.cubeA.pose.p - self.goal_position) <= self.cube_half_size
    is_robot_static = np.max(np.abs(self.robot.qvel)) <= 0.2

    # If the cube is placed at the goal and the robot is static, return a high reward
    if cube_at_goal and is_robot_static:
        reward += 2.25
        return reward

    # reaching reward, encourages the robot to reach the cube
    gripper_pos = self.robot.ee_pose.p
    obj_pos = self.cubeA.pose.p
    dist_to_obj = np.linalg.norm(gripper_pos - obj_pos)
    reaching_reward = 1 - np.tanh(5 * dist_to_obj)
    reward += reaching_reward

    # grasp reward, encourages the robot to grasp the cube
    is_grasped = self.robot.check_grasp(self.cubeA, max_angle=30)
    if is_grasped:
        reward += 0.25

    # placement reward, encourages the robot to place the cube at the goal
    if is_grasped:
        dist_to_goal = np.linalg.norm(self.cubeA.pose.p - self.goal_position)
        placement_reward = 1 - np.tanh(5 * dist_to_goal)
        reward += placement_reward

    # regularization term on robot's action
    action_reg = -np.sum(np.square(action)) / len(action)
    reward += 0.1 * action_reg

    return reward
```

*Oracle* (expert-written) reward function of MANISKILL2 Pick Cube task:

```
import numpy as np

def compute_dense_reward(self, action):
    reward = 0.0

    is_obj_placed = np.linalg.norm(self.goal_position - self.cubeA.pose.p) <= 0.025
    is_robot_static = np.max(np.abs(self.robot.qvel)) <= 0.2

    success = is_obj_placed and is_robot_static

    if success:
        reward += 5
        return reward

    tcp_to_obj_pos = self.cubeA.pose.p - self.robot.ee_pose.p
    tcp_to_obj_dist = np.linalg.norm(tcp_to_obj_pos)
```

```
    reaching_reward = 1 - np.tanh(5 * tcp_to_obj_dist)
    reward += reaching_reward

    is_grasped = self.robot.check_grasp(self.cubeA, max_angle=30)
    reward += 1 if is_grasped else 0.0

    if is_grasped:
        obj_to_goal_dist = np.linalg.norm(self.goal_position - self.cubeA.pose.p)
        place_reward = 1 - np.tanh(5 * obj_to_goal_dist)
        reward += place_reward

    return reward
```

Sample of *zero-shot* generated reward function on MANISKILL2 Turn Faucet task:

```
import numpy as np
from scipy.spatial.distance import cdist

def compute_dense_reward(self, action):
    # Here I define the reward and penalty weights for different aspects of the task
    handle_reach_weight = 0.1
    grasp_handle_weight = 0.2
    rotation_weight = 0.7
    action_penalty_weight = 0.01

    # Here I define the grasp success reward
    grasp_success_reward = 0.1

    # Here I compute the distance between the robot's end effector and the faucet handle
    handle_pcd = self.faucet.handle.get_world_pcd()
    lfinger_cords = np.array([self.robot.lfinger.pose.p, ])
    rfinger_cords = np.array([self.robot.rfinger.pose.p, ])
    dist_lfinger_handle = cdist(lfinger_cords, handle_pcd).min(-1)[0]
    dist_rfinger_handle = cdist(rfinger_cords, handle_pcd).min(-1)[0]
    dist_handle_reach = max(dist_lfinger_handle, dist_rfinger_handle)
    handle_reach_reward = - handle_reach_weight * dist_handle_reach

    # Here I check if the robot has successfully grasped the faucet handle
    grasp_handle = self.robot.check_grasp(self.faucet.handle)
    grasp_handle_reward = grasp_handle_weight * grasp_handle if grasp_handle else 0

    # Here I calculate the rotation reward based on the difference between the current and
    # target joint position
    rotation_diff = max(0, self.faucet.handle.target_qpos - self.faucet.handle.qpos)
    rotation_reward = - rotation_weight * rotation_diff

    # Here I calculate the penalty for the robot's action
    action_penalty = - action_penalty_weight*np.linalg.norm(action)

    # The total reward is the sum of all the individual rewards and penalties
    reward = handle_reach_reward + grasp_handle_reward + rotation_reward + action_penalty

    if grasp_handle and rotation_diff == 0:
        reward += grasp_success_reward

    return reward
```

Sample of *few-shot* generated reward function on MANISKILL2 Turn Faucet task:

```
import numpy as np
from scipy.spatial.distance import cdist

def compute_dense_reward(self, action):
    reward = 0.0

    # check if the handle is turned on
    is_handle_turned_on = self.faucet.handle.qpos >= self.faucet.handle.target_qpos
    # check if the robot is static
    is_robot_static = np.max(np.abs(self.robot.qvel)) <= 0.2

    success = is_handle_turned_on and is_robot_static

    if success:
        reward += 5
        return reward

    # calculate the reward for reaching the handle
    tcp_to_handle_pos = self.faucet.handle.pose.p - self.robot.ee_pose.p
    tcp_to_handle_dist = np.linalg.norm(tcp_to_handle_pos)
    reaching_reward = 1 - np.tanh(5 * tcp_to_handle_dist)
```

```
    reward += reaching_reward

    # calculate the reward for successfully grasping the handle
    is_grasped = self.robot.check_grasp(self.faucet.handle, max_angle=30)
    reward += 1 if is_grasped else 0.0

    if is_grasped:
        # reward for the robot turning the handle
        handle_to_target_dist = self.faucet.handle.target_qpos - self.faucet.handle.qpos
        turning_reward = 1 - np.tanh(5 * handle_to_target_dist)
        reward += turning_reward

    # penalize actions that could potentially damage the robot
    action_penalty = np.sum(np.square(action))
    reward -= action_penalty * 0.1

    return reward
```

*Oracle* (expert-written) generated reward function on MANISKILL2 Turn Faucet task:

```
def compute_dense_reward(self, action):
    reward = 0.0

    angle_diff = self.faucet.handle.target_qpos - self.faucet.handle.qpos
    success = angle_diff < 0

    if success:
        return 10.0

    """Compute the distance between the tap and robot fingers."""
    pcd = self.faucet.handle.get_world_pcd()
    pcd1 = self.robot.lfinger.get_world_pcd()
    pcd2 = self.robot.rfinger.get_world_pcd()

    distance1 = cdist(pcd, pcd1)
    distance2 = cdist(pcd, pcd2)

    distance = min(distance1.min(), distance2.min())

    reward += 1 - np.tanh(distance * 5.0)

    turn_reward_1 = 3 * (1 - np.tanh(max(angle_diff, 0) * 2.0))
    reward += turn_reward_1

    delta_angle = angle_diff - self.last_angle_diff
    if angle_diff > 0:
        turn_reward_2 = -np.tanh(delta_angle * 2)
    else:
        turn_reward_2 = np.tanh(delta_angle * 2)
    turn_reward_2 *= 5
    reward += turn_reward_2

    self.last_angle_diff = angle_diff

    return reward
```

Sample of *zero-shot* generated reward function on MANISKILL2 Open Cabinet Door task:

```
def compute_dense_reward(self, action):
    # Define weights for different parts of the reward
    w_distance = 0.4
    w_goal = 0.4
    w_action = 0.2

    # Calculate distance between robot's gripper and the cabinet handle
    handle_pcd = self.cabinet.handle.get_world_pcd()
    ee_cords = self.robot.get_ee_coords().reshape(-1, 3)
    distance = cdist(ee_cords, handle_pcd).min()
    distance_reward = -w_distance * distance  # Negative reward since we want to minimize the distance

    # Calculate the difference between current state of cabinet drawer and its goal state
    # Positive reward since we want to maximize the qpos
    goal_diff = self.cabinet.handle.qpos - self.cabinet.handle.target_qpos
    goal_reward = w_goal * goal_diff

    # Add regularization of robot's action, penalize large actions
    action_reward = -w_action * np.linalg.norm(action)

    # Check if the target drawer is static, if so, give a large positive reward
    if self.cabinet.handle.check_static():
```

```
        static_reward = 1.0
    else:
        static_reward = 0.0

    # Combine different parts of the reward
    reward = distance_reward + goal_reward + action_reward + static_reward

    return reward
```

Sample of *few-shot* generated reward function on MANISKILL2 Open Cabinet Door task:

```
from scipy.spatial import distance as sdist

def compute_dense_reward(self, action: np.ndarray) -> float:
    reward = 0

    # Compute distance from end-effector to cabinet door handle
    ee_coords = self.robot.get_ee_coords()  # [2, 3]
    handle_pcd = self.cabinet.handle.get_world_pcd()  # [N, 3]

    # EE approach handle
    dist_ee_to_handle = sdist.cdist(ee_coords, handle_pcd)  # [2, N]
    dist_ee_to_handle = dist_ee_to_handle.min(1)  # [2]
    dist_ee_to_handle = dist_ee_to_handle.mean()
    log_dist_ee_to_handle = np.log(dist_ee_to_handle + 1e-5)
    reward += -dist_ee_to_handle - np.clip(log_dist_ee_to_handle, -10, 0)

    # Penalize action
    # Assume action is relative and normalized.
    action_norm = np.linalg.norm(action)
    reward -= action_norm * 1e-6

    # Cabinet door position
    cabinet_door_pos = self.cabinet.handle.qpos
    target_door_pos = self.cabinet.handle.target_qpos

    # Stage reward
    stage_reward = -10
    if cabinet_door_pos > target_door_pos:
        # Cabinet door is opened
        stage_reward += 5
    else:
        # Encourage the robot to continue moving the cabinet door
        stage_reward += (cabinet_door_pos - target_door_pos) * 2

    reward = reward + stage_reward
    return reward
```

*Oracle* (expert-written) generated reward function on MANISKILL2 Open Cabinet Door task:

```
def compute_dense_reward(self, action):
    reward = 0.0

    # ---------------------------------------------------------------------- #
    # The end-effector should be close to the target pose
    # ---------------------------------------------------------------------- #
    handle_pose = self.cabinet.handle.pose
    ee_pose = self.robot.ee_pose

    # Position
    ee_coords = self.robot.get_ee_coords()  # [2, 10, 3]
    handle_pcd = self.cabinet.handle.get_world_pcd()

    disp_ee_to_handle = sdist.cdist(ee_coords.reshape(-1, 3), handle_pcd)
    dist_ee_to_handle = disp_ee_to_handle.reshape(2, -1).min(-1)  # [2]
    reward_ee_to_handle = -dist_ee_to_handle.mean() * 2
    reward += reward_ee_to_handle

    # Encourage grasping the handle
    ee_center_at_world = ee_coords.mean(0)  # [10, 3]
    ee_center_at_handle = transform_points(
        handle_pose.inv().to_transformation_matrix(), ee_center_at_world
    )

    dist_ee_center_to_handle = self.cabinet.handle.local_sdf(ee_center_at_handle)

    dist_ee_center_to_handle = dist_ee_center_to_handle.max()
    reward_ee_center_to_handle = (
        clip_and_normalize(dist_ee_center_to_handle, -0.01, 4e-3) - 1
    )
```

```
        reward += reward_ee_center_to_handle

        # Rotation
        target_grasp_poses = self.cabinet.handle.target_grasp_poses
        target_grasp_poses = [handle_pose * x for x in target_grasp_poses]
        angles_ee_to_grasp_poses = [angle_distance(ee_pose, x) for x in target_grasp_poses]
        ee_rot_reward = -min(angles_ee_to_grasp_poses) / np.pi * 3
        reward += ee_rot_reward

        # ------------------------------------------------------------------------- #
        # Stage reward
        # ------------------------------------------------------------------------- #
        coeff_qvel = 1.5  # joint velocity
        coeff_qpos = 0.5  # joint position distance
        stage_reward = -5 - (coeff_qvel + coeff_qpos)

        link_qpos = self.cabinet.handle.qpos
        link_qvel = self.cabinet.handle.qvel
        link_vel_norm = np.linalg.norm(self.cabinet.handle.velocity)
        link_ang_vel_norm = np.linalg.norm(self.cabinet.handle.angular_velocity)

        ee_close_to_handle = (dist_ee_to_handle.max() <= 0.01 and dist_ee_center_to_handle > 0)
        if ee_close_to_handle:
            stage_reward += 0.5

            # Distance between current and target joint positions
            reward_qpos = (clip_and_normalize(link_qpos, 0, self.cabinet.handle.target_qpos) * coeff_qpos)
            reward += reward_qpos

            if link_qpos < self.cabinet.handle.target_qpos:
                # Encourage positive joint velocity to increase joint position
                reward_qvel = clip_and_normalize(link_qvel, -0.1, 0.5) * coeff_qvel
                reward += reward_qvel
            else:
                # Add coeff_qvel for smooth transition of stagess
                stage_reward += 2 + coeff_qvel
                reward_static = -(link_vel_norm + link_ang_vel_norm * 0.5)
                reward += reward_static

                if link_vel_norm <= 0.1 and link_ang_vel_norm <= 1:
                    stage_reward += 1

        reward += stage_reward
        return reward
```

Sample of *zero-shot* generated reward function on MANISKILL2 Open Cabinet Drawer task:

```
import numpy as np
from scipy.spatial.distance import cdist

def compute_dense_reward(self, action):
    # Define weights for different parts of the reward
    w_distance = 0.4
    w_goal = 0.4
    w_action = 0.2

    # Calculate distance between robot's gripper and the cabinet handle
    handle_pcd = self.cabinet.handle.get_world_pcd()
    ee_cords = self.robot.get_ee_coords().reshape(-1, 3)
    distance = cdist(ee_cords, handle_pcd).min()
    distance_reward = -w_distance * distance  # Negative reward since we want to minimize the distance

    # Calculate the difference between current state of cabinet drawer and its goal state
    # Positive reward since we want to maximize the qpos
    goal_diff = self.cabinet.handle.qpos - self.cabinet.handle.target_qpos
    goal_reward = w_goal * goal_diff

    # Add regularization of robot's action, penalize large actions
    action_reward = -w_action * np.linalg.norm(action)

    # Check if the target drawer is static, if so, give a large positive reward
    if self.cabinet.handle.check_static():
        static_reward = 1.0
    else:
        static_reward = 0.0

    # Combine different parts of the reward
    reward = distance_reward + goal_reward + action_reward + static_reward

    return reward
```

Sample of *few-shot* generated reward function on MANISKILL2 Open Cabinet Drawer task:

```python
from scipy.spatial import distance as sdist
import numpy as np

def compute_dense_reward(self, action: np.ndarray):
    reward = 0

    # Compute distance between end-effector and cabinet handle surface
    ee_coords = np.array(self.robot.get_ee_coords())  # [2, 10, 3]
    handle_pcd = self.cabinet.handle.get_world_pcd()  # [N, 3]

    # EE approach handle
    dist_ees_to_handle = sdist.cdist(ee_coords.reshape(-1, 3), handle_pcd)  # [20, N]
    dist_ees_to_handle = dist_ees_to_handle.min(0)  # [N]
    dist_ee_to_handle = dist_ees_to_handle.mean()
    log_dist_ee_to_handle = np.log(dist_ee_to_handle + 1e-5)
    reward += -dist_ee_to_handle - np.clip(log_dist_ee_to_handle, -10, 0)

    # Penalize action
    # Assume action is relative and normalized.
    action_norm = np.linalg.norm(action)
    reward -= action_norm * 1e-6

    # Encourage qpos change
    qpos_change = self.cabinet.handle.qpos - self.cabinet.handle.target_qpos
    reward += qpos_change * 0.1

    # Penalize the velocity of cabinet and handle
    handle_vel_norm = np.linalg.norm(self.cabinet.handle.velocity)
    reward -= handle_vel_norm * 0.01
    cabinet_vel_norm = np.linalg.norm(self.cabinet.velocity)
    reward -= cabinet_vel_norm * 0.01

    # Stage reward
    stage_reward = -10
    if dist_ee_to_handle < 0.1:
        # EE is close to handle
        stage_reward += 2
        if self.cabinet.handle.qpos >= self.cabinet.handle.target_qpos:
            # The drawer is open
            stage_reward += 8
    reward += stage_reward

    return reward
```

*Oracle* (expert-written) generated reward function on MANISKILL2 Open Cabinet Drawer task:

```python
def compute_dense_reward(self, action):
    reward = 0.0

    # -------------------------------------------------------------------------- #
    # The end-effector should be close to the target pose
    # -------------------------------------------------------------------------- #
    handle_pose = self.cabinet.handle.pose
    ee_pose = self.robot.ee_pose

    # Position
    ee_coords = self.robot.get_ee_coords()  # [2, 10, 3]
    handle_pcd = self.cabinet.handle.get_world_pcd()

    disp_ee_to_handle = sdist.cdist(ee_coords.reshape(-1, 3), handle_pcd)
    dist_ee_to_handle = disp_ee_to_handle.reshape(2, -1).min(-1)  # [2]
    reward_ee_to_handle = -dist_ee_to_handle.mean() * 2
    reward += reward_ee_to_handle

    # Encourage grasping the handle
    ee_center_at_world = ee_coords.mean(0)  # [10, 3]
    ee_center_at_handle = transform_points(
        handle_pose.inv().to_transformation_matrix(), ee_center_at_world
    )

    dist_ee_center_to_handle = self.cabinet.handle.local_sdf(ee_center_at_handle)

    dist_ee_center_to_handle = dist_ee_center_to_handle.max()
    reward_ee_center_to_handle = (
        clip_and_normalize(dist_ee_center_to_handle, -0.01, 4e-3) - 1
    )
    reward += reward_ee_center_to_handle
```

```
    # Rotation
    target_grasp_poses = self.cabinet.handle.target_grasp_poses
    target_grasp_poses = [handle_pose * x for x in target_grasp_poses]
    angles_ee_to_grasp_poses = [angle_distance(ee_pose, x) for x in target_grasp_poses]
    ee_rot_reward = -min(angles_ee_to_grasp_poses) / np.pi * 3
    reward += ee_rot_reward

    # ------------------------------------------------------------------------- #
    # Stage reward
    # ------------------------------------------------------------------------- #
    coeff_qvel = 1.5  # joint velocity
    coeff_qpos = 0.5  # joint position distance
    stage_reward = -5 - (coeff_qvel + coeff_qpos)

    link_qpos = self.cabinet.handle.qpos
    link_qvel = self.cabinet.handle.qvel
    link_vel_norm = np.linalg.norm(self.cabinet.handle.velocity)
    link_ang_vel_norm = np.linalg.norm(self.cabinet.handle.angular_velocity)

    ee_close_to_handle = (dist_ee_to_handle.max() <= 0.01 and dist_ee_center_to_handle > 0)
    if ee_close_to_handle:
        stage_reward += 0.5

        # Distance between current and target joint positions
        reward_qpos = (clip_and_normalize(link_qpos, 0, self.cabinet.handle.target_qpos) * coeff_qpos)
        reward += reward_qpos

        if link_qpos < self.cabinet.handle.target_qpos:
            # Encourage positive joint velocity to increase joint position
            reward_qvel = clip_and_normalize(link_qvel, -0.1, 0.5) * coeff_qvel
            reward += reward_qvel
        else:
            # Add coeff_qvel for smooth transition of stagess
            stage_reward += 2 + coeff_qvel
            reward_static = -(link_vel_norm + link_ang_vel_norm * 0.5)
            reward += reward_static

            if link_vel_norm <= 0.1 and link_ang_vel_norm <= 1:
                stage_reward += 1

    reward += stage_reward
    return reward
```

Sample of *zero-shot* generated reward function on MANISKILL2 Push Chair task:

```
import numpy as np
from scipy.spatial.distance import cdist

def compute_dense_reward(self, action) -> float:
    # Stage 1: Encourage the robot to move towards the chair
    # Get the distance between the robot's gripper and the chair
    gripper_coords = self.robot.get_ee_coords()
    chair_pcd = self.chair.get_pcd()
    dist_to_chair = cdist(gripper_coords, chair_pcd).min(-1).mean()

    # Get the difference between the chair's current and target position
    chair_to_target_dist = np.linalg.norm(self.chair.pose.p[:2] - self.target_xy)

    # The smaller the distance, the larger the reward
    reward_dist = -dist_to_chair
    # The closer the chair is to the target, the larger the reward
    reward_target = -chair_to_target_dist

    # Stage 2: Encourage the robot to push the chair towards the target location
    # Get the velocity of the chair
    chair_vel = self.chair.velocity[:2]
    # The faster the chair moves towards the target, the larger the reward
    reward_vel = np.dot(chair_vel, (self.target_xy - self.chair.pose.p[:2])) / \
            (np.linalg.norm(chair_vel) * chair_to_target_dist)

    # Stage 3: Prevent the chair from falling over
    # Calculate the tilt angle of the chair
    z_axis_chair = self.chair.pose.to_transformation_matrix()[:3, 2]
    chair_tilt = np.arccos(z_axis_chair[2])
    # The smaller the tilt, the larger the reward
    reward_tilt = -chair_tilt

    # Regularization of the robot's action
    reward_reg = -np.square(action).sum()

    # Weights for each stage
```

```
    w_dist = 0.2
    w_target = 0.2
    w_vel = 0.3
    w_tilt = 0.2
    w_reg = 0.1

    # Final reward
    reward = w_dist * reward_dist + w_target * reward_target + w_vel * reward_vel +
            w_tilt * reward_tilt + w_reg * reward_reg

    return reward
```

Sample of *few-shot* generated reward function on MANISKILL2 Push Chair task:

```
import numpy as np
import scipy.spatial.distance as sdist

def compute_dense_reward(self, action):
    reward = -20.0

    actor = self.chair
    ee_coords = np.array(self.robot.get_ee_coords())
    ee_mids = np.array([ee_coords[:2].mean(0), ee_coords[2:].mean(0)])
    chair_pcd = self.chair.get_pcd()

    # EE approach chair
    dist_ees_to_chair = sdist.cdist(ee_coords, chair_pcd)  # [4, N]
    dist_ees_to_chair = dist_ees_to_chair.min(1)   # [4]
    dist_ee_to_chair = dist_ees_to_chair.mean()
    log_dist_ee_to_chair = np.log(dist_ee_to_chair + 1e-5)
    reward += - dist_ee_to_chair - np.clip(log_dist_ee_to_chair, -10, 0)

    # Penalize action
    action_norm = np.linalg.norm(action)
    reward -= action_norm * 1e-6

    # Keep chair standing
    z_axis_world = np.array([0, 0, 1])
    z_axis_chair = self.chair.pose.to_transformation_matrix()[:3, 2]
    chair_tilt = np.arccos(z_axis_chair[2])
    log_chair_tilt = np.log(chair_tilt + 1e-5)
    reward += -chair_tilt * 0.2

    # Chair velocity
    chair_vel = actor.velocity
    chair_vel_norm = np.linalg.norm(chair_vel)
    disp_chair_to_target = self.chair.pose.p[:2] - self.target_xy
    chair_vel_dir = sdist.cosine(chair_vel[:2], disp_chair_to_target)
    chair_ang_vel_norm = np.linalg.norm(actor.angular_velocity)

    # Stage reward
    stage_reward = 0

    dist_chair_to_target = np.linalg.norm(self.chair.pose.p[:2] - self.target_xy)

    if dist_ee_to_chair < 0.2:
        stage_reward += 2
        if dist_chair_to_target <= 0.3:
            stage_reward += 2
            reward += (np.exp(-chair_vel_norm * 10) * 2)
            if chair_vel_norm <= 0.1 and chair_ang_vel_norm <= 0.2:
                stage_reward += 2
                if chair_tilt <= 0.1 * np.pi:
                    stage_reward += 2
        else:
            reward_vel = (chair_vel_dir - 1) * chair_vel_norm
            reward += np.clip(1 - np.exp(-reward_vel), -1, np.inf)*2 - dist_chair_to_target*2

    if chair_tilt > 0.4 * np.pi:
        stage_reward -= 2

    reward = reward + stage_reward
    return reward
```

*Oracle* (expert-written) generated reward function on MANISKILL2 Push Chair task:

```
def compute_dense_reward(self, action: np.ndarray):
    reward = 0

    # Compute distance between end-effectors and chair surface
```

```
    ee_coords = np.array(self.robot.get_ee_coords())  # [4, 3]
    chair_pcd = self.chair.get_world_pcd()  # [N, 3]

    # EE approach chair
    dist_ees_to_chair = sdist.cdist(ee_coords, chair_pcd)  # [4, N]
    dist_ees_to_chair = dist_ees_to_chair.min(1)  # [4]
    dist_ee_to_chair = dist_ees_to_chair.mean()
    log_dist_ee_to_chair = np.log(dist_ee_to_chair + 1e-5)
    reward += -dist_ee_to_chair - np.clip(log_dist_ee_to_chair, -10, 0)

    # Keep chair standing
    # z-axis of chair should be upward
    z_axis_chair = self.chair.pose.to_transformation_matrix()[:3, 2]
    chair_tilt = np.arccos(z_axis_chair[2])
    reward += -chair_tilt * 0.2

    # Penalize action
    # Assume action is relative and normalized.
    action_norm = np.linalg.norm(action)
    reward -= action_norm * 1e-6

    # Chair velocity
    chair_vel = self.chair.velocity[:2]
    chair_vel_norm = np.linalg.norm(chair_vel)
    disp_chair_to_target = self.chair.pose.p[:2] - self.target_xy
    cos_chair_vel_to_target = sdist.cosine(disp_chair_to_target, chair_vel)
    chair_ang_vel_norm = np.linalg.norm(self.chair.angular_velocity)

    # Stage reward
    stage_reward = -10
    disp_chair_to_target = self.chair.pose.p[:2] - self.target_xy
    dist_chair_to_target = np.linalg.norm(disp_chair_to_target)

    if chair_tilt < 0.2 * np.pi:
        # Chair is standing
        if dist_ee_to_chair < 0.1:
            # EE is close to chair
            stage_reward += 2
            if dist_chair_to_target <= 0.15:
                # Chair is close to target
                stage_reward += 2
                # Try to keep chair static
                reward += np.exp(-chair_vel_norm * 10) * 2
                if chair_vel_norm <= 0.1 and chair_ang_vel_norm <= 0.2:
                    stage_reward += 2
            else:
                # Try to increase velocity along direction to the target
                # Compute directional velocity
                x = (1 - cos_chair_vel_to_target) * chair_vel_norm
                reward += max(-1, 1 - np.exp(x)) * 2 - dist_chair_to_target * 2
    else:
        stage_reward = -5

    reward = reward + stage_reward
    return reward
```

## D.2 NOVEL LOCOMOTION RESULTS

Sample of *zero-shot* generated reward function on MUJOCO Hopper Front Flip task:

```
def compute_dense_reward(self, action) -> float:

    # Reward for forward movement
    reward_x_velocity = self.hopper.top.velocity_x * 1.0

    # Reward for maintaining a low body height to prepare for flip
    reward_low_height = -abs(self.hopper.top.position_z - 0.5) * 0.5

    # Reward for fast positive angular velocity for top joint
    reward_top_joint_velocity = self.hopper.top_joint.angular_velocity * 1.0 if \
                                self.hopper.top_joint.angular_velocity > 0 else 0.0

    # Reward for fast negative angular velocity for leg and foot joints
    reward_leg_foot_velocity = -self.hopper.leg_joint.angular_velocity * 0.5 if \
                                self.hopper.leg_joint.angular_velocity < 0 else 0.0
    reward_leg_foot_velocity += -self.hopper.foot_joint.angular_velocity * 0.5 if \
                                 self.hopper.foot_joint.angular_velocity < 0 else 0.0

    # Penalty for not completing a flip (i.e., top_joint angle is not 360 degree or 2*pi radian)
```

```
    penalty_incomplete_flip = -abs(self.hopper.top_joint.angle - 2*math.pi) * 0.1

    # Summing up the rewards and penalties
    reward = reward_x_velocity + reward_low_height + reward_top_joint_velocity + \
            reward_leg_foot_velocity + penalty_incomplete_flip

    return reward
```

Sample of *zero-shot* generated reward function on MUJOCO Hopper Back Flip task:

```
def compute_dense_reward(self, action) -> float:
    # Reward for rotating around the foot joint, which means a flip is happening
    reward_flip = abs(self.hopper.foot_joint.angular_velocity)

    # Penalty for touching the ground with something other than the foot
    touching_ground_penalty = 0
    if self.hopper.top.position_z < 1.25 or self.hopper.thigh_joint.angle < 0 or \
                                    self.hopper.leg_joint.angle < 0:
        touching_ground_penalty = -100

    # Penalty for not maintaining a reasonable speed
    speed_penalty = 0
    if abs(self.hopper.top.velocity_x)<0.1 or abs(self.hopper.foot_joint.angular_velocity)<0.1:
        speed_penalty = -50

    # Reward for moving in the x-coordinate direction
    reward_x = self.hopper.top.velocity_x
    if reward_x < 0:
        reward_x *= 10

    # Total reward
    reward = reward_flip + touching_ground_penalty + speed_penalty + reward_x

    return reward
```

Sample of *zero-shot* generated reward function on MUJOCO Ant Lie Down task:

```
def compute_dense_reward(self, action) -> float:
    # Define the weight constants
    W_pos = 1.0  # weight for the torso's z-coordinate position
    W_vel = 0.1  # weight for the torso's velocity
    W_act = 0.01 # weight for the action (torques applied)

    # Calculate position reward (the closer the torso's z position is to 0, the higher the reward)
    pos_reward = -W_pos * self.ant.torso.position_z

    # Calculate the velocity penalty (the slower the torso's velocity, the higher the reward)
    vel_penalty = -W_vel * (self.ant.torso.velocity_x**2 + \
                self.ant.torso.velocity_y**2 + self.ant.torso.velocity_z**2)

    # Calculate the action penalty (the smaller the action values, the higher the reward)
    action_penalty = -W_act * np.sum(np.array(action)**2)

    # Combine the rewards and penalties to compute the final reward
    reward = pos_reward + vel_penalty + action_penalty

    return reward
```

Sample of *zero-shot* generated reward function on MUJOCO Ant Wave Leg task:

```
def compute_dense_reward(self, action) -> float:
    # Define the initial reward
    reward = 0.0

    # Keeping the ant in place
    # We penalize any significant movement in x, y, and z position
    reward -= abs(self.ant.torso.velocity_x) * 0.1
    reward -= abs(self.ant.torso.velocity_y) * 0.1
    reward -= abs(self.ant.torso.position_z - 0.75) * 1.0

    # Keeping the ant's torso from rotating
    # We penalize any significant angular velocity
    reward -= abs(self.ant.torso.angular_velocity_x) * 0.1
    reward -= abs(self.ant.torso.angular_velocity_y) * 0.1
    reward -= abs(self.ant.torso.angular_velocity_z) * 0.1

    # Waving the right front leg
    # We give positive reward for movement in the front right leg
    reward += abs(self.ant.front_right_leg_joint.angular_velocity) * 0.2
    reward += abs(self.ant.front_right_ankle_joint.angular_velocity) * 0.2
```

```
    # Keeping other legs stationary
    # We penalize movement in the other legs
    for leg_joint, ankle_joint in [
        (self.ant.front_left_leg_joint, self.ant.front_left_ankle_joint),
        (self.ant.back_left_leg_joint, self.ant.back_left_ankle_joint),
        (self.ant.back_right_leg_joint, self.ant.back_right_ankle_joint)]:

        reward -= abs(leg_joint.angular_velocity) * 0.1
        reward -= abs(ankle_joint.angular_velocity) * 0.1

    return reward
```

## D.3   INTERACTIVE RESULTS

For the interactive generation results on MANISKILL2 Stack Cube task, we provide the traces of the whole round which contains the initial generated reward function, the human feedback and the revised reward function in *git diff* format.

Instruction: *Pick up cube A and place it on cube B. The task is finished when cube A is on top of cube B stably (i.e. cube A is static) and isn't grasped by the gripper.*

Initial generated reward function:

```
import numpy as np

def compute_dense_reward(self, action):
    reward = 0.0

    # Check if cube A is on top of cube B and whether it is stable
    is_obj_on_target = np.linalg.norm(self.cubeA.pose.p - (self.cubeB.pose.p + \
                        np.array([0, 0, self.cube_half_size*2]))) <= 0.025
    is_obj_static = self.cubeA.check_static()
    is_grasped = self.robot.check_grasp(self.cubeA)

    success = is_obj_on_target and is_obj_static and not is_grasped

    if success:
        reward += 5
        return reward

    # Compute distance from the robot's end-effector to cube A
    tcp_to_obj_pos = self.cubeA.pose.p - self.robot.ee_pose.p
    tcp_to_obj_dist = np.linalg.norm(tcp_to_obj_pos)
    reaching_reward = 1 - np.tanh(5 * tcp_to_obj_dist)
    reward += reaching_reward

    # Check if the robot has grasped cube A
    reward += 1 if is_grasped else 0.0

    if is_grasped:
        # Compute distance from cube A to the top of cube B
        obj_to_goal_pos = self.cubeB.pose.p + np.array([0, 0, self.cube_half_size*2]) - \
                        self.cubeA.pose.p
        obj_to_goal_dist = np.linalg.norm(obj_to_goal_pos)
        place_reward = 1 - np.tanh(5 * obj_to_goal_dist)
        reward += place_reward

    return reward
```

Human description: *Now the robot only picks up cube A, stacks it onto cube B, but does not release cube A afterwards.*

Human feedback for improvement: *The robot should release cube A after stacking it onto cube B.*

Revised reward function (Interaction 1):

```
import numpy as np

def compute_dense_reward(self, action):
    reward = 0.0

    # Check if cube A is on top of cube B and whether it is stable
    is_obj_on_target = np.linalg.norm(self.cubeA.pose.p - (self.cubeB.pose.p + \
                        np.array([0, 0, self.cube_half_size*2]))) <= 0.025
    is_obj_static = self.cubeA.check_static()
    is_grasped = self.robot.check_grasp(self.cubeA)
```

```
-   success = is_obj_on_target and is_obj_static and not is_grasped
+   success = is_obj_on_target and is_obj_static

    if success:
        reward += 5
-       return reward
+       if not is_grasped:
+           reward += 5
+           return reward

    # Compute distance from the robot's end-effector to cube A
    tcp_to_obj_pos = self.cubeA.pose.p - self.robot.ee_pose.p
    tcp_to_obj_dist = np.linalg.norm(tcp_to_obj_pos)
    reaching_reward = 1 - np.tanh(5 * tcp_to_obj_dist)
    reward += reaching_reward

    # Check if the robot has grasped cube A
    reward += 1 if is_grasped else 0.0

    if is_grasped:
        # Compute distance from cube A to the top of cube B
        obj_to_goal_pos = self.cubeB.pose.p + np.array([0, 0, self.cube_half_size*2]) - \
                                    self.cubeA.pose.p
        obj_to_goal_dist = np.linalg.norm(obj_to_goal_pos)
        place_reward = 1 - np.tanh(5 * obj_to_goal_dist)
        reward += place_reward

+       if is_obj_on_target:
+           # Encourage the robot to release the grasp when cube A is on top of cube B
+           reward += 3 if not is_grasped else -3

    return reward
```

Human description: *Now the robot sometimes completes the task successfully, but sometimes fails.*

Human feedback for improvement: *Please write denser reward to complete the task more stably.*

Revised reward function (Interaction 2):

```
import numpy as np

def compute_dense_reward(self, action):
    reward = 0.0

    # Check if cube A is on top of cube B and whether it is stable
    is_obj_on_target = np.linalg.norm(self.cubeA.pose.p - (self.cubeB.pose.p + \
                    np.array([0, 0, self.cube_half_size*2]))) <= 0.025
    is_obj_static = self.cubeA.check_static()
    is_grasped = self.robot.check_grasp(self.cubeA)
+   gripper_openness = self.robot.gripper_openness

    success = is_obj_on_target and is_obj_static

    if success:
        reward += 5
        if not is_grasped:
            reward += 5
            return reward

    # Compute distance from the robot's end-effector to cube A
    tcp_to_obj_pos = self.cubeA.pose.p - self.robot.ee_pose.p
    tcp_to_obj_dist = np.linalg.norm(tcp_to_obj_pos)
    reaching_reward = 1 - np.tanh(5 * tcp_to_obj_dist)
    reward += reaching_reward

    # Check if the robot has grasped cube A
    reward += 1 if is_grasped else 0.0

    if is_grasped:
        # Compute distance from cube A to the top of cube B
        obj_to_goal_pos = self.cubeB.pose.p + np.array([0, 0, self.cube_half_size*2]) - \
                        self.cubeA.pose.p
        obj_to_goal_dist = np.linalg.norm(obj_to_goal_pos)
```

```
        place_reward = 1 - np.tanh(5 * obj_to_goal_dist)
        reward += place_reward

-       if is_obj_on_target:
-            # Encourage the robot to release the grasp when cube A is on top of cube B
-            reward += 3 if not is_grasped else -3
+       if is_obj_on_target and is_obj_static:
+            # Encourage robot to release grasp when cube A is on top of cube B and stable
+            release_reward = 3 if not is_grasped and gripper_openness > 0.5 else -3
+            reward += release_reward
+       else:
+            # Encourage the robot to keep the grasp if it's not on target
+            holding_reward = 1 if is_grasped else -1
+            reward += holding_reward
+  else:
+       if not is_obj_on_target:
+            # Encourage the robot to move closer to the target when not grasping
+            tcp_to_target_pos = self.cubeB.pose.p + np.array([0, 0, self.cube_half_size*2])
+                                      - self.robot.ee_pose.p
+            tcp_to_target_dist = np.linalg.norm(tcp_to_target_pos)
+            target_reward = 1 - np.tanh(5 * tcp_to_target_dist)
+            reward += target_reward

    return reward
```

# E    ADDITIONAL ANALYSIS OF EFFECTIVENESS OVER PREVIOUS WORK

As detailed in Section 4.1 of our work, modifications to the baseline method of Yu et al. (2023) (L2R) enable adaptation to only two tasks of the MANISKILL2 environment, while L2R is not suited for tasks that involve complexly surfaced objects, which cannot be adequately described by a singular point. To further comprehensively evaluate the significance of the shaped dense reward structure, particularly a staged reward necessitating Python's *if-else* statements, we introduce an additional baseline. This baseline adapts the oracle expert-written reward codes to mirror the L2R's format (the sum of a set of individual terms) by removing all *if-else* statements and only keeping functional reward terms, while disregarding the original L2R's inability to utilize point clouds for distance calculations.

As illustrated in Figure 8, *Oracle-L2R* attains comparable outcomes to the *zero-shot* setting for three relatively straightforward and short-horizon tasks: Lift Cube, Pick Cube, and Turn Faucet. However, for the remaining three tasks, which are comparatively more difficult, the modified *Oracle-L2R* fails to address the challenges, underscoring the necessity of shaped and staged dense rewards.

# F    EXPERIMENTS ON OPEN-SOURCE LANGUAGE MODELS

We conduct experiments using open-source Large Language Models (LLMs) as our foundational model to ensure reproducibility and accessibility. Specifically, we employ the instruction-tuned with human feedback version of Llama-2 (Touvron et al., 2023)[5] and Code-Llama (Rozière et al., 2023)[6] as representative strongest open-source models, establishing them as the baseline.

Experiments are conducted on MetaWorld tasks within a zero-shot setting, adhering to the methodology described in the main body of our work. The resulting learning curves of the policies trained using code generated from these two open-source models, in conjunction with GPT-4, are illustrated in Figure 9.

---

[5]https://huggingface.co/meta-llama/Llama-2-7b-chat-hf
[6]https://huggingface.co/codellama/CodeLlama-34b-Instruct-hf

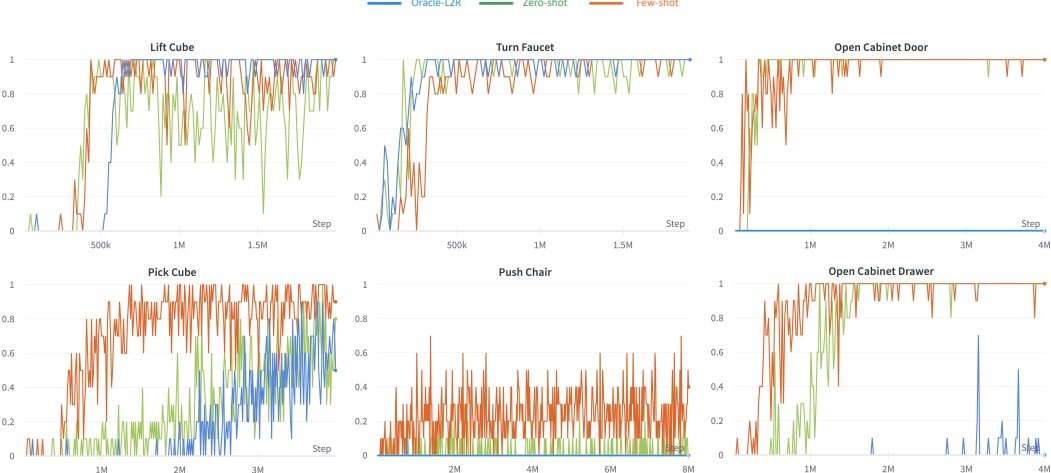

Figure 8: Learning curves on MANISKILL2 compared to **oracle** Yu et al. (2023) settings, measured by task success rate. *Oracle-L2R* means that we modify the expert-written reward function provided by the environment into the format of Yu et al. (2023), which is only a sum of different reward terms, without *if-else* statements and other possible components of Python; *zero-shot* and *few-shot* stands for the reward function is generated by TEXT2REWARD w.o and w. retrieving examples from expert-written rewards functions examples for prompting.

The learning curves illustrate the evolution of policy performance on MetaWorld tasks under a zero-shot reward generation setting. It is evident that GPT-4 consistently outperforms the other models across most tasks, achieving higher success rates and displaying more stable learning progressions. The Code-Llama model exhibits moderate success, with notable variability in tasks such as 'Window Close' and 'Door Unlock'. Llama-2, while lagging behind in tasks like 'Drawer Open' and 'Door Close', shows promise in 'Button Press' with a learning curve that approaches GPT-4's performance. The shaded regions indicate standard deviations, suggesting that GPT-4's policy training is not only more successful on average but also more reliable across different random seeds. These results underscore the sophistication of GPT-4's code generation in producing effective reward functions for policy learning in a zero-shot setting, and the effectiveness of proof-of-the-concept exploration in this direction using it.

It is noteworthy that, despite many models approaching the performance of ChatGPT and GPT-4 on existing benchmarks (Cobbe et al., 2021; Hendrycks et al., 2021; Chen et al., 2021), there remains a significant gap in the reward function generation task, suggesting that this problem could potentially serve as one of the benchmark tests for large language models and that the benchmark has considerable room for improvement. Enhancements could simultaneously benefit both the natural language processing and reinforcement learning communities.

# G    ADDITIONAL RESULTS

Due to page limitations, we include in this section additional learning curves on METAWORLD benchmark (Figure 10), as well as additional image samples from the MANISKILL2 and METAWORLD environments (Figure 11 and Figure 12). Figure 11b shows a failure case on the dual-arm mobile manipulation task Push Chair. Here *Oracle* means the expert-written reward function provided by the environment; *zero-shot* and *few-shot* stands for the reward function is generated by TEXT2REWARD w. and w.o. retrieving examples from expert-written rewards functions examples for prompting.

## G.1    ERROR ANALYSIS ON GENERATED FUNCTION

We conduct an error analysis on the generated reward functions. We manually go over 100 reward function examples each generated by the zero-shot and few-shot prompting on 10 different tasks of MANISKILL2, where each task has 10 different reward codes. These reward functions are generated

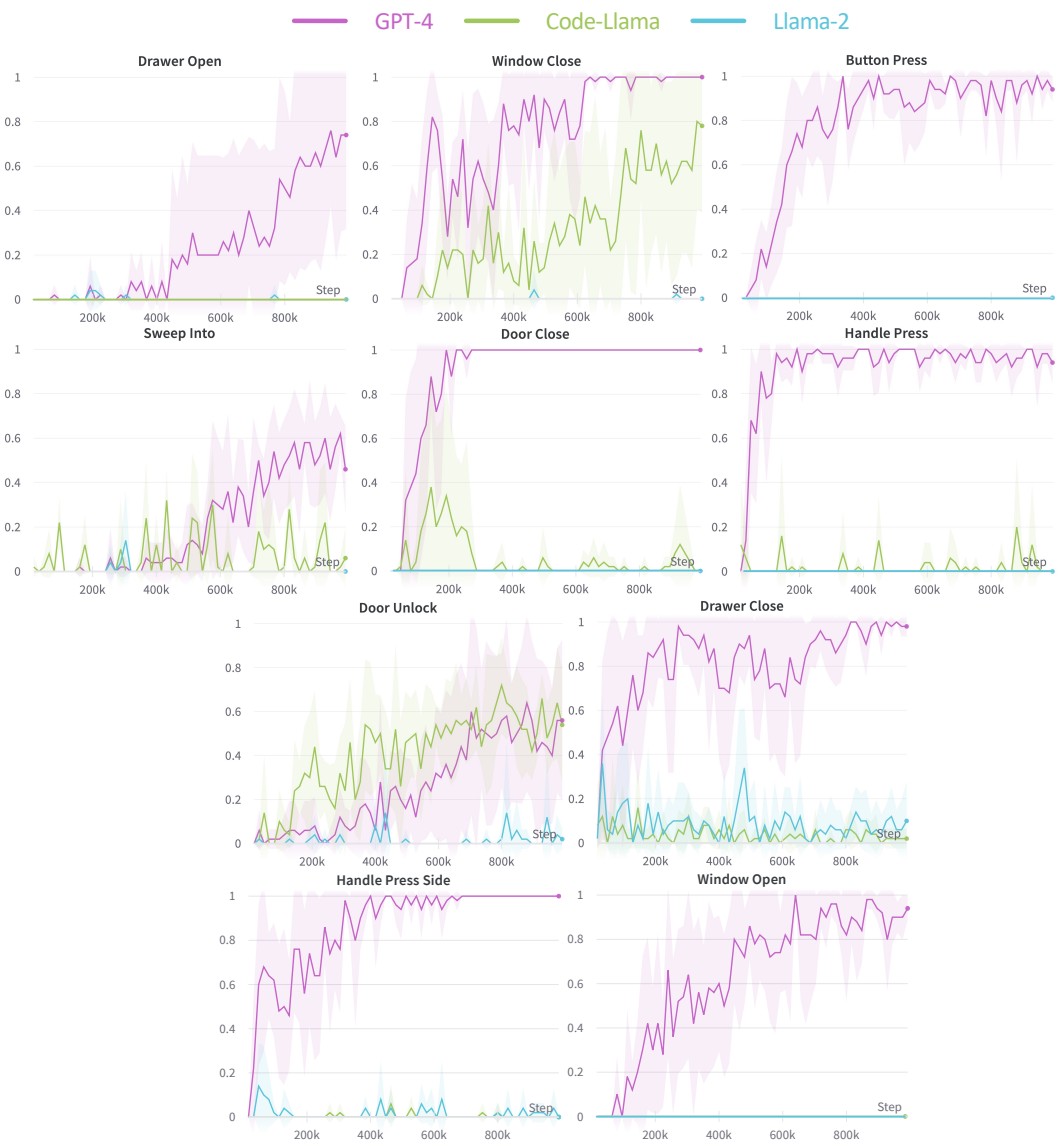

Figure 9: Learning curves on METAWORLD under zero-shot reward generation setting, measured by success rate, using the reward functions generated by different language models. The solid line represents the mean success rate, while the shaded regions correspond to the standard deviation, both calculated across five different random seeds.

specifically for our error analysis and with no execution feedback to LLMs. We classify them into 4 error types: Class attributes misuse; Attributes hallucination (referring to attributes that do not exist); Syntax/shape error; Wrong package. Table 7 shows that the overall error rate is around 10%. Within these error samples, 30% of the errors are caused by the code syntax or shape mismatch, the rest are introduced during grounding from background knowledge context to choose the existing yet right one, indicating there is still space to improve on understanding how to choose the right function and attributes for TEXT2REWARD direction, especially for code generation community.

## H  LIMITATIONS AND FUTURE WORK

Our work demonstrates the effectiveness of generating dense reward functions for RL. We focus on the code-based reward format, which gives us high interpretability. However, the symbolic space may not cover all aspects of the reward. Furthermore, our method also assumes that the perception

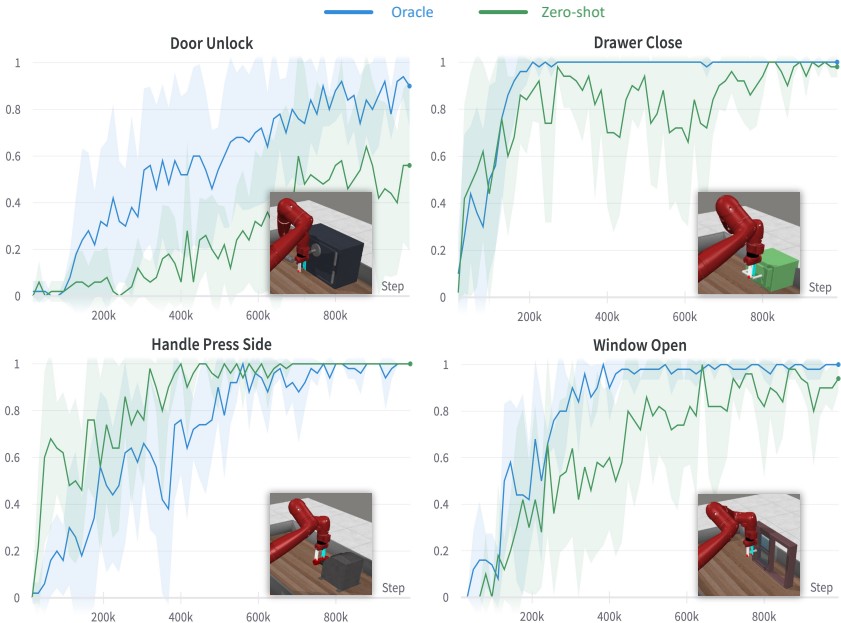

Figure 10: Additional learning curves on METAWORLD, measured by success rate. The solid line represents the mean success rate, while the shaded regions correspond to the standard deviation, both calculated across five different random seeds.

Table 7: Error distribution across 100 generated reward code on MANISKILL2.

| Type | Description | Zero-shot | Few-shot |
|---|---|---|---|
| Class attribute misuse | Use other classes' attribute wrongly | 6% | 4% |
| Attribute hallucination | Invent nonexistent attribute | 3% | 2% |
| Syntax/shape error | Incorrect program grammar or shape mismatch | 3% | 3% |
| Wrong package | Import incorrect package function | 1% | 1% |
| Correct | Execute correctly without error | 87% | 90% |

is already done with other off-the-shelf components. Future works may consider the combination of code-based and neural network-based reward design that combines both symbolic reasoning and perception. Utilizing the knowledge derived from LLMs in creating such models showcases promising prospects and could be advantageous in several scenarios (Wulfmeier et al., 2016; Finn et al., 2016; Christiano et al., 2017; Lee et al., 2021).

Although our main method is simple but effective, it still has room for improvement by designing methods to generate better reward functions, possibly leading to higher success rates and the ability to tackle more complex tasks.

At present, our test cases primarily concentrate on robotics tasks, specifically manipulation and loco-motion, to illustrate this approach. In the future, this research may find broader applications in various reinforcement learning related domains, including gaming (Brockman et al., 2016; Schrittwieser et al., 2020; Zhong et al., 2021; Fan et al., 2022), web navigation (Shi et al., 2017; Zhou et al., 2023), household management (Puig et al., 2018; Shridhar et al., 2020a;b).

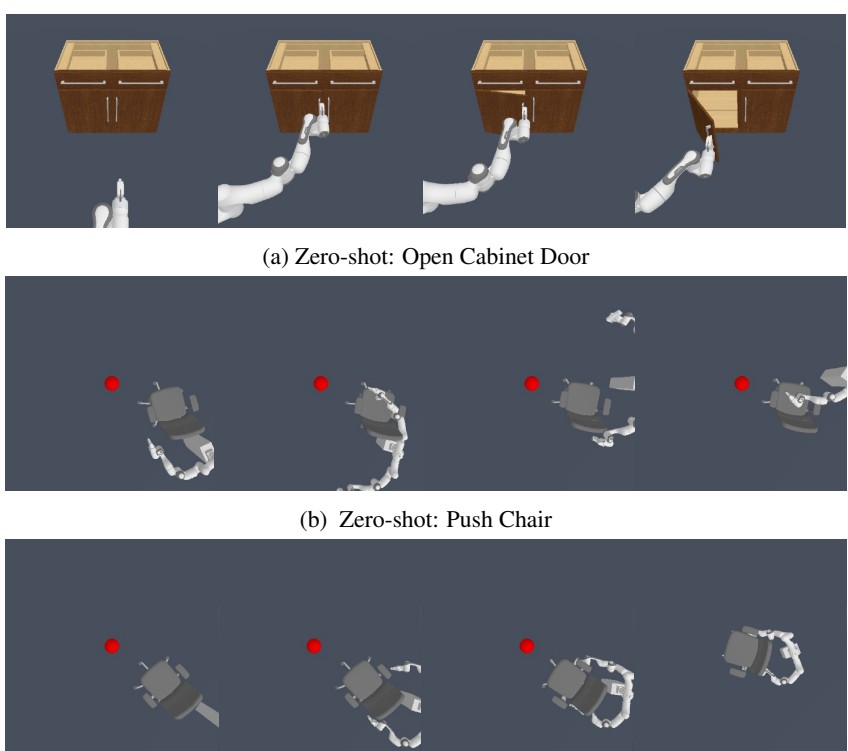

(a) Zero-shot: Open Cabinet Door

(b) Zero-shot: Push Chair

(c) Few-shot: Push Chair

Figure 11: Additional image samples from MANISKILL2. (a) shows a successful case of zero-shot TEXT2REWARD for the mobile manipulation task OpenCabinetDoor. (b) shows a failure case of zero-shot generation for the dual-arm mobile manipulation task PushChair. (c) shows that few-shot generation can stably solve the PushChair task even though zero-shot can not.

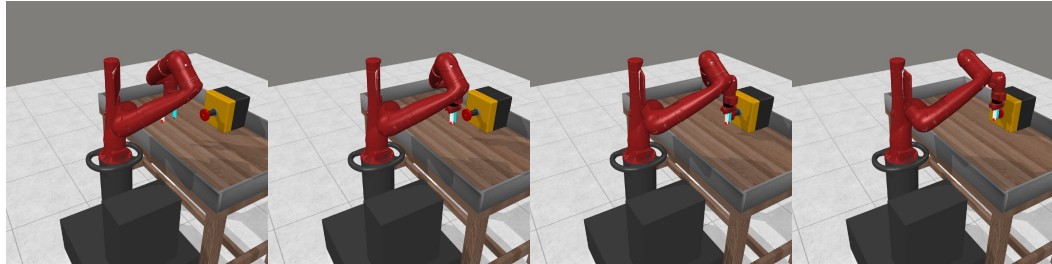

(a) Zero-shot: Button Press

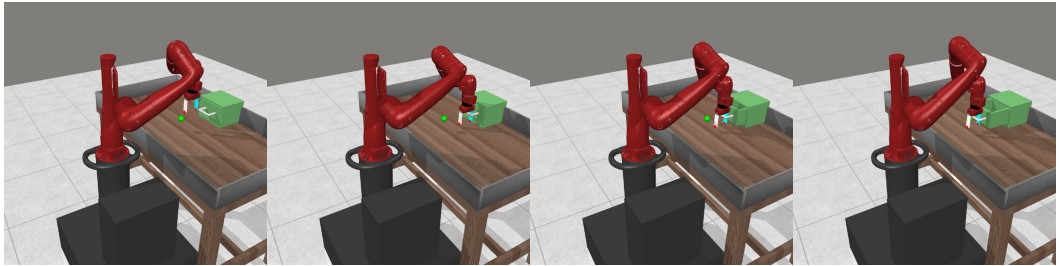

(b) Zero-shot: Drawer Open

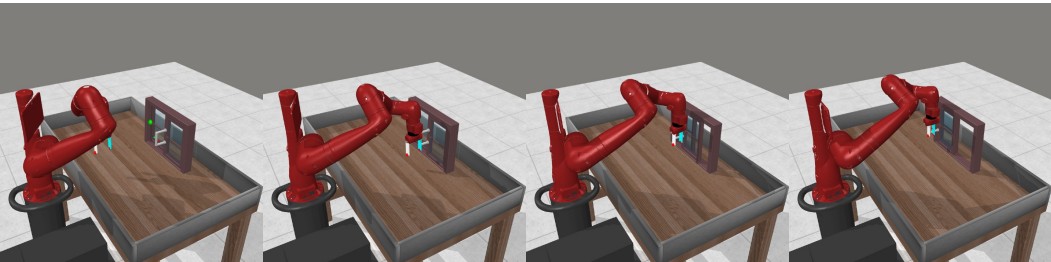

(c) Zero-shot: Window Open

Figure 12: Additional image samples from METAWORLD. All three rollouts show that zero-shot TEXT2REWARD can stably solve these tasks.

