# OpenReview forum: "Text2Reward: Reward Shaping with Language Models for Reinforcement Learning"
_ICLR.cc/2024/Conference — ICLR 2024 spotlight_

### Official Review · Reviewer_bPzM · 2023-10-26

**Soundness:** 2 fair
**Presentation:** 3 good
**Contribution:** 2 fair
**Rating:** 6
**Confidence:** 5

**Summary:**

The authors propose a method for leveraging language models to write code computing reward functions for simulated reinforcement learning tasks, given a textual description of the task.

Their method makes use of an expert provided environment “abstraction” (a specification) written as a sequence of pythonic class declarations, and combines it with a human provided textual instruction describing the desired trainable task. This is used to prompt GPT-4 into generating python code for computing rewards. Optionally, the prompt given to the language model can be augmented with few-shot examples of expert and previously-generated reward functions, retrieved from a database using embedding similarity between desired and stored instructions.

Within the proposed framework, it is also possible to iteratively improve the generated reward function by having a human provide written and descriptive feedback given a video from a policy learnt with the current generated reward. Such feedback is introduced in the language model prompt and used to generate a new version of the reward function code.
In a series of experiments over a variety of simulated environments, the authors demonstrate the effectiveness of the approach, while generally showing few-shot and iterative prompting to improve results. Overall, generated reward functions appear to be similar in performance to expert provided rewards.

**Strengths:**

The paper addresses the important problem of dense reward specification in reinforcement learning, using a method based on zero-shot and few-shot prompting of a language model, something not shown before except for concurrent work **[1]**.

The authors show that their method is competitive with human expert reward specification in a broad series of experiments. They moreover demonstrate one such policy to transfer successfully to real world execution.

The authors also include an analysis of the failure modes of reward specification, specifically for the cases in which the generated reward function code leads to a python runtime error.

**[1]** Ma et al., Eureka: Human-Level Reward Design via Coding Large Language Models, https://eureka-research.github.io/

**Weaknesses:**

The paper essentially proposes prompting techniques for code generation with pre-trained language models accessible via API, specifically GPT-4. Despite its interesting conclusion and results, it consists at most of an interesting observational study over the capabilities of GPT-4, as the prompting techniques appear straightforward, fundamentally easy to execute manually by any user of the GPT-4 online api platform in a process of trial and error for reward design. I would not consider this a problem worthy of rejection per se, as similar “prompting techniques” results have been published before in machine learning venues, such as the famous “chain of thought” prompting technique. Still, it is my opinion that this limits the significance of the work, as GPT-4 simply fills in the shoes of an expert reinforcement learning reward coder on well known simulation environments (while “chain of thought” prompting was an innovation in how to query a model for more general purpose NLP benchmarks).

More importantly, to my understanding, no truly novel environments are tackled in this paper. For most of the shown tasks, GPT-4 can reasonably be expected to draw from training data containing countless reward function specifications from those environments, if not reward functions for the desired task itself.

Moreover, the reliance of the paper on GPT-4 makes the results inherently irreproducible, as OpenAI does not have a policy of indefinitely supporting api access to specific snapshot versions of their models (specifically, GPT-4-0314 will be deprecated on June 13th 2024 at the earliest).

**Questions:**

* Would it be possible to show how this method generalizes to environments that are sure not to have been included in the training data of the model? (e.g: a simple simulated environment designed specifically for this paper)
* Did you try your method on open source or otherwise publicly available models that would allow for a snapshot of the results to be preserved? If so, how do they compare to GPT-4?

Edit: I raised the score according to my latest comment.

---

> ### Author Response · Authors · 2023-11-16
>
> Dear Reviewer bPzM,
>
> Thank you for your kind feedback and constructive comments. And thanks for your recognition of our work in addressing the important problem of dense reward specification in reinforcement learning, using language model prompting.
>
> We have revised our manuscript and are addressing the specific questions in the following reply, including:
> - discussion on data leakage and generalization ability
> - incorporation of experiments on Llama-2 and Code-Llama on 10 tasks of MetaWorld; discussion and improvement on reproducibility
>
> Please let us know if you have any further questions and we can provide any additional clarifications to help finalize your assessment and rating of our paper.
>
> ----
>
> ### **W1: The significance of our work**
> **A:** We appreciate the reviewer's perspective. However, we respectfully disagree it is limited to the significance of the method this work proposed. To clarify, we want to emphasize the promise of this work for different communities (RL, Robotics, and NLP) as follows:
>
> For the **reinforcement learning community**, it is promising to use LLMs to automate policy learning by generating reward functions that are difficult to write down by hand. It is not only an important application scenario for replacing human experts in writing rewards functions but also a novel way that has never been shown before to pour LLMs’ knowledge and abilities into the RL for robotics and control, as well as games. Our work shows promising initial results for further exploration in the future (as shown in the main experiments of Figure 2).
>
> For the **robotics community**, one of the most important things in the field of robotics is how to learn low-level motor skills at large scales efficiently, scalably, and automatically (or in a self-supervised manner) [1][2]. We believe that by leveraging LLMs, our method (automatic text-to-reward) takes a solid step toward automatically learning large-scale motor skills.
>
> For the **natural language processing community**, our work proposes text-to-reward as a hard task for code generation and shows a huge gap in performance between the most advanced model and ones from the open-source community (as shown in the newly added experiments on Llama-2 and Code-Llama in Appendix F), showing the potential of improving in this aspect for people in the NLP community.
>
> Overall, our method could be an important direction in the future as a novel way to pour the knowledge learned from LLMs into general RL, low-level robotics skills learning and control, which are not the focus of previous studies that focus on high-level planning, and the method could get more effective with the growth of LLMs.
>
> [1] Yufei Wang, Zhou Xian, Feng Chen, Tsun-Hsuan Wang, Yian Wang, Zackory Erickson, David Held, Chuang Gan, RoboGen: Towards Unleashing Infinite Data for Automated Robot Learning via Generative Simulation https://arxiv.org/abs/2311.01455
>
> [2] Open X-Embodiment: Robotic Learning Datasets and RT-X Models, https://arxiv.org/abs/2310.08864
>
> ----
>
> ### **W1 & Q1: Generalization to environments not included in GPT-4's training data.**
> **A:** Thanks for your question! We understand your concern regarding the potential data leakage of environment and reward code in GPT-4 and other LLMs, which might lead one to assume that the model's ability to generate high-quality code stems from exposure to "golden environments" and "golden examples."
>
> However, we would like to clarify **three sources of evidence** from our paper and previous work:
> 1. Evidence from the input side (environment and prompts) and output side (generated reward function) of our proposed method for reward function generation;
> 2. Evidence from the locomotion novel skills experiments (including backflip, frontflip, wave legs) part in our paper;
> 3. Evidence from previous research showed the generalization capability on code generation problems of new domains that are not included in the training set.
>
> **1. Evidence from the input side (prompt) and output side (generated reward function) of our proposed method for reward function generation.**
>
> As much as you suggest using a simple, purpose-designed simulated environment, we would like to clarify that the Python classes and properties for our prompts were crafted by our team and differ significantly from the environment's original code. We then map the generated code to the environment's original code using a one-to-one deterministic mapping.

---

> > ### Author Response · Authors · 2023-11-16
> >
> > **From the input end,** one piece of evidence, as shown in Appendix C, is that the prompts we provided to LLMs like GPT-4 for our T2R method are **derived from expertly abstracted class Python representations, with specific attributes and function names coined by us, not exact snippets from the ManiSkill2 or MetaWorld GitHub repositories**. This rules out the possibility of the language model plagiarizing directly through context memory since previous work [1] has attempted to perturb instructions used for code generation and found significant performance drops.
> >
> > On the other hand, we also have preliminary results that entering direct commands like "generate a reward function for crawler" or "pick the cube" into GPT-4, or inputting the ManiSkill2 environment code context into GPT-4 to let it complete the `compute_dense_reward` section, none of which were successful. This not only demonstrates the inadequacy of mere memorization but also validates the effectiveness of our method.
> >
> > **From the output end,** another piece of evidence is shown in Appendix D.1 of our paper, where we compare instances of code generated by LLM through zero-shot and few-shot prompting with the oracle code example on the ManiSkill2 Pick Cube task.
> >
> > Below is the code generated by zero-shot prompting GPT-4 using the T2R method:
> > ```python
> > import numpy as np
> >
> > def compute_dense_reward(self, action):
> >     # Normalize action
> >     action = np.clip(action, -1, 1)
> >
> >     # Calculate distance between gripper and cube
> >     gripper_pos = self.robot.ee_pose.p
> >     cube_pos = self.cubeA.pose.p
> >     dist_gripper_cube = np.linalg.norm(gripper_pos - cube_pos)
> >
> >     # Calculate distance between cube and goal
> >     goal_pos = self.goal_position dist_cube_goal = np.linalg.norm(goal_pos - cube_pos)
> >
> >     # Check if the robot is grasping the cube
> >     grasping_cube = self.robot.check_grasp(self.cubeA)
> >
> >     # Define reward components
> >     reward_dist_gripper_cube = -1.0 * dist_gripper_cube
> >     reward_dist_cube_goal = -1.0 * dist_cube_goal
> >     reward_grasping_cube = 1.0 if grasping_cube else -1.0
> >
> >     # Define weights for reward components
> >     weight_dist_gripper_cube = 0.3
> >     weight_dist_cube_goal = 0.5
> >     weight_grasping_cube = 0.2
> >
> >     # Calculate total reward
> >     reward = weight_dist_gripper_cube * reward_dist_gripper_cube \
> >                 + weight_dist_cube_goal * reward_dist_cube_goal \
> >                 + weight_grasping_cube * reward_grasping_cube
> >
> >     # Regularization on action
> >     reward -= 0.01 * (action ** 2).sum()
> >     return reward
> > ```
> >
> > And here is the oracle code (from the experts of the ManiSkill team):
> > ```
> > import numpy as np
> >
> > def compute_dense_reward(self, action):
> >     reward = 0.0
> >
> >     is_obj_placed = np.linalg.norm(self.goal_position - self.cubeA.pose.p) <= 0.025
> >     is_robot_static = np.max(np.abs(self.robot.qvel)) <= 0.2
> >
> >     success = is_obj_placed and is_robot_static
> >
> >     if success:
> >         reward += 5
> >         return reward
> >
> >     tcp_to_obj_pos = self.cubeA.pose.p - self.robot.ee_pose.p
> >     tcp_to_obj_dist = np.linalg.norm(tcp_to_obj_pos)
> >     reaching_reward = 1 - np.tanh(5 * tcp_to_obj_dist)
> >     reward += reaching_reward
> >
> >     is_grasped = self.robot.check_grasp(self.cubeA, max_angle=30)
> >     reward += 1 if is_grasped else 0.0
> >
> >     if is_grasped:
> >         obj_to_goal_dist = np.linalg.norm(self.goal_position - self.cubeA.pose.p)
> >         place_reward = 1 - np.tanh(5 * obj_to_goal_dist)
> >         reward += place_reward
> >
> >     return reward
> > ```
> >
> > If the process were reliant on memory replication, these two code segments would be very similar. However, here the styles and approaches of the reward are quite different, yet they ultimately achieve the same effect. Thus, we wouldn’t say the "leakage" argument negates the efficacy of the T2R method.
> >
> > **2. Evidence from the locomotion novel skills experiments (including backflip, frontflip, wave legs) part in our paper**
> >
> > We did experiments on learning novel skills in MuJoCo, as shown in Table 1 and Figure 4, T2R generates reward codes under our created new instructions for novel skills. We also show the generated reward function codes in Appendix D.2.

---

> > > ### Author Response · Authors · 2023-11-16
> > >
> > > **3. Evidence from previous research showed the generalization capability on code generation problems of new domains that are not included in the training set**
> > >
> > > Previous research has shown the generalization capability and great performance on code generation problems that are not included in the training set of GPT-4 by conducting:
> > >
> > >  - contamination study [2]: According to Appendix C (Contamination on professional and academic exams), Appendix D (Contamination on academic benchmarks), Table 9, Table 10, and Table 11, the GPT-4 team from OpenAI trained different versions of GPT-4 w. and w.o. the content from the evaluation set and conclude “contamination overall has very little effect on the reported results.”.
> > >  - testing on fresh coding questions after GPT-4’s pretraining period [3]. They construct a benchmark of 100 LeetCode problems posted after October 8th, 2022, which is after GPT-4’s pretraining period and show that GPT-4 significantly outperforms the other models and is comparable to human performance on these fresh questions.
> > >
> > > [1] Yuhang Lai, Chengxi Li, Yiming Wang, Tianyi Zhang, Ruiqi Zhong, Luke Zettlemoyer, Scott Wen-tau Yih, Daniel Fried, Sida Wang, Tao Yu. DS-1000: A Natural and Reliable Benchmark for Data Science Code Generation https://arxiv.org/abs/2211.11501
> > >
> > > [2] OpenAI, GPT-4 Technical Report
> > > https://arxiv.org/abs/2303.08774
> > >
> > > [3] Sébastien Bubeck, Varun Chandrasekaran, Ronen Eldan, Johannes Gehrke, Eric Horvitz, Ece Kamar, Peter Lee, Yin Tat Lee, Yuanzhi Li, Scott Lundberg, Harsha Nori, Hamid Palangi, Marco Tulio Ribeiro, Yi Zhang. Sparks of Artificial General Intelligence: Early experiments with GPT-4
> > > https://arxiv.org/abs/2303.12712
> > >
> > > ----
> > >
> > > ### **W1 & Q2: Tests with open source or publicly available models for reproducibility.**
> > > **A:** Thanks for your suggestion! As you mentioned, it is a common practice to use the advanced GPT models from OpenAI or close source model PaLM from Google for novel applications and studies in previous work (Chain-of-Thought[1], Zero-shot planner[2], SayCan[3], ReAct[4]). For the reproducibility, we used open-source algorithms and environments and provided full reproducible prompts and code.
> > >
> > > Following your suggestion, we also add experiments in the application of our method using open-source or publicly available models on Llama-2 and Code-Llama (Llama-2 further pretrained on code corpus) on 10 tasks of MetaWorld. We include these findings in our revised manuscript and provide a more comprehensive evaluation of our method's capabilities across different models in Appendix F. The results show that the open-source models still have a huge gap with the most advanced models in the difficult task of reward function generation and reflect the necessity of using the most advanced one to demonstrate the possibility of our method.
> > >
> > > [1] Jason Wei, Xuezhi Wang, Dale Schuurmans, Maarten Bosma, Brian Ichter, Fei Xia, Ed Chi, Quoc Le, Denny Zhou. Chain-of-Thought Prompting Elicits Reasoning in Large Language Models.
> > > https://arxiv.org/abs/2201.11903
> > >
> > > [2] Wenlong Huang, Pieter Abbeel, Deepak Pathak, Igor Mordatch. Language Models as Zero-Shot Planners: Extracting Actionable Knowledge for Embodied Agents.
> > > https://arxiv.org/abs/2201.07207
> > >
> > > [3] Michael Ahn, Anthony Brohan, Noah Brown, et al,. Do As I Can, Not As I Say: Grounding Language in Robotic Affordances
> > > https://arxiv.org/abs/2204.01691
> > >
> > > [4] Shunyu Yao, Jeffrey Zhao, Dian Yu, Nan Du, Izhak Shafran, Karthik Narasimhan, Yuan Cao. ReAct: Synergizing Reasoning and Acting in Language Models.
> > > https://arxiv.org/abs/2210.03629

---

> > > > ### Comment · Reviewer_bPzM · 2023-11-21
> > > > **Response to Authors**
> > > >
> > > > Dear Authors,
> > > > I thank you for your long and detailed reply. After considering your points, I elected to raise my score to a 6.
> > > > Your explanation of your use of abstract Python class representations, together with your other points related to generalization, convinced me that complete memorization of reward functions should not be an issue with your approach.
> > > > As for your comparison with open source models, the negative result is worrying insofar as the success of the techniques described in your paper explicitly depends on the scale and (more importantly) the data distribution used to train GPT-4 specifically, which is not known to us. This is why I will not raise my score more than this.

---

> > > > > ### Author Response · Authors · 2023-11-21
> > > > >
> > > > > We are glad you have an updated evaluation of our paper! We agree that the performance of Text2Reward depends on ability of LLM, especially the ability of combine code generation, planning, and commonsense (the latter two are important for writing staged rewards). We hope Text2Reward, along with other concurrent works, can be a useful resource to evaluate and improve LLM’s ability to automate reward shaping. We will open-source our code and prompts to facilitate this process.
> > > > >
> > > > > Thank you for your constructive feedback! Feel free to put any other thoughts before the discussion closes.

---

### Official Review · Reviewer_VQaY · 2023-10-27

**Soundness:** 3 good
**Presentation:** 4 excellent
**Contribution:** 3 good
**Rating:** 8
**Confidence:** 4

**Summary:**

The authors present a method to use LLMs to write dense reward functions to train policies from natural language instructions.

**Strengths:**

************************************************************Evaluation Comprehensiveness:************************************************************ The authors utilized 17 tasks in ManiSkill2 and Metaworld, 6 in Mujoco, and also a few real world experiments — this is quite comprehensive compared to the average ICLR RL paper.

****************************Experiments:****************************

- Real world experiments are always nice!
- Human feedback experiments are also great and demonstrate that on certain tasks the method can improve the policy learning performance — something that isn’t always straightforward even with humans redesigning reward functions to try to improve polichy learning

******************Results:****************** I think results are solid compared to the oracle.

**********Motivation:********** The paper motivation is nice — reducing human effort in scaling policy training by using LLMs for python/numpy reward function design.

**Weaknesses:**

************************************************************Contribution over prior work:************************************************************ I’m not too convinced on the major **********technical********** contribution over Yu et al. 2023 (Language to Rewards for Skill Synthesis). Compared to that paper, the main claimed novelty is dense vs sparse reward and the use of standard python code: “Different from recent work….our free-form dense reward code has a wider coverage of tasks and can use established coding packages.” But that paper also uses very pythonic code, furthermore utilizes sparse reward mainly due to using MPC. I’m not too convinced that the pythonic → python and MPC → RL are large technical contributions on their own. This should be clarified more specifically/clearly in the paper if there is another technical contribution over Yu 2023, and if not, then is one of the main reasons for my score.

************************Experiments:************************

- The authors should compare against Yu 2023, especially if claiming their dense free-form reward + use of established coding packages can result in superior performance. The comparison isn’t exactly 1-1 given the claims, but currently there is no comparison to any baseline to contextualize the performance of the method. In fact, I think a comparison with Yu 2023 + RL would be fairest, as Yu 2023 likely can use RL instead of MPC without change.
- Open-source LLMs: Utilizing closed-source LLMs has obvious downsides, e.g., reproducibility (API backend can change at any time) and access to academic researchers (cost per token vs able to be used on a standard GPU setup). It would be beneficial to the community to demonstrate some results with some smaller open source models like LLaMa-2.

**************************Minor Issues:**************************

- 4.1: Appendex → Appendix
- I think it’d be nice to have a few small examples in the **********main paper********** of generated reward functions (not full things, just a few lines). This makes the experiment section more readable without needing to jump around to the appendix.

**Questions:**

Putting it here because this is not a “weakness” as this paper just came out: Eureka has recently come out (https://eureka-research.github.io/), and as concurent work, what do the authors think they contribute compared to Eureka?

---

> ### Author Response · Authors · 2023-11-16
>
> Dear Reviewer VQaY,
>
> Thank you for your appreciation and detailed evaluation of our work. We are glad you think the evaluation of our method is comprehensive and your appreciation of the real-robot experiments and human-feedback findings. We have revised our manuscript and addressed several points you mentioned.
>
> We have revised our manuscript and are addressing the specific questions in the following reply, including:
> 1. discussion on technical contributions
> 2. incorporation of 2 baselines on 6 tasks of ManiSkill2 to justify technical contributions
> 3. incorporation of experiments of LLama-2 and Code Llama on 10 tasks of MetaWorld; add discussion and improvement on reproducibility
>
> Please let us know if you have any further questions and we can provide any additional clarifications to help finalize your assessment and rating of our paper.
>
> -----
>
> ### **W1: Technical contribution compared to Yu et al. 2023**
> **A:** Thanks for your question and advice! We understand your primary concern pertains to the comparison with the L2R paper by Yu et al., 2023, in terms of technical contributions. We wish to clarify there are fundamental differences between L2R and our method (T2R).
>
> Our method's technical contribution extends beyond L2R in **three key aspects**:
> 1. **L2R can only use limited reward terms v.s. T2R can use any reward term**
> 2. **L2R can only use a sum of reward terms v.s. T2R can write stage and dense rewards such as if-else statements and can incorporate any other Python feature**
> 3. **L2R is model-based and needs huge expert effort (for building the MPC model and writing reward term templates and even weights) v.s. T2R is model-free and can easily be transferred to new environments and real-world application**
>
> To demonstrate these, we have incorporated **two new experiments**:
> 1. We introduce a new baseline of "L2R prompt + RL" in the main body of our paper and test the baseline on 6 ManiSkill2 tasks. **"L2R prompt + RL" was only deployable on 2 tasks due to the limitation of its reward terms (contribution#1), falling short in the remaining 4 tasks, underscoring the limitations of the task coverage of L2R.**
> 2. We introduce an upgraded version of L2R --- "oracle-sparse-reward + RL" --- in Appendix E and test this baseline on 6 ManiSkill2 tasks to further demonstrate the importance of stage and dense reward. **While "oracle-sparse-reward + RL" yielded results comparable to the zero-shot setting for 3 simpler tasks, it completely can not work (i.e. success rate ≈ 0) on the remaining 3 more challenging tasks, underscoring the limitations of sparse rewards (contribution#2).**
>
> For more detailed explanation of these **three differences**:
> 1. **L2R can only use limited reward terms v.s. T2R can use any reward term:**
>    1. The use of Model Predictive Control (MPC) within the L2R framework restricts the reward term to a twice-differentiable norm with its minimum at zero, typically the L2 distance.
>    2. This presents a limitation when dealing with the ManiSkill2 benchmark, which uses a point cloud to depict complex, articulated rigid bodies with intricate surfaces, offering a more realistic representation of real-world scenarios. L2R’s inability to go beyond the L2 distance measure between vectors restricts its representation of such complex geometries.
>    3. In contrast, our method's flexibility supports a broader spectrum of coding packages and styles, thus enhancing its adaptability across various environments and tasks.
>    4. Our experimental results further highlight this distinction: when testing the baseline on 6 ManiSkill2 tasks, “L2R prompt + RL” was only deployable in 2 tasks due to the limitation of its reward terms, falling short in the remaining 4 tasks.

---

> ### Author Response · Authors · 2023-11-16
>
> 2. **L2R can only use a sum of reward terms v.s. T2R can write stage and dense rewards such as if-else statements:**
>    1. Our use of dense rewards, as opposed to sparse rewards, allows for a more nuanced and continuous evaluation of agent behaviors, crucial for complex tasks that involve many steps.
>    2. For example, the stack cube task needs "pick up cubeA, place it on cubeB, release cubeA and make sure cubeA is static after releasing". However, L2R will lead to a contradiction between the reward term that encourages "pick up cubeA" and "release cubeA" when writing them in a simple summation expression.
>    3. To show the necessity of dense rewards, we trained RL with oracle-sparse-reward on ManiSkill2 as an additional experiment in Appendix E. This baseline adapts the oracle expert-written reward codes to mirror the L2R's format (the sum of a set of individual terms) by removing all if-else statements and only keeping functional reward terms, while disregarding the original L2R's inability to utilize point clouds for distance calculations. While "oracle-sparse-reward + RL" yielded results comparable to the zero-shot setting for 3 simpler tasks, it completely can not work (i.e. success rate ≈ 0) on the remaining 3 more challenging tasks, underscoring the limitations of sparse rewards.
> 3. **L2R is model-based and needs additional expert effort v.s. T2R is model-free and can easily be transferred to new environments and real-world application:**
>    1. The efficacy of the L2R approach is significantly contingent on the performance of Model Predictive Control (MPC). While it is a well-accepted notion that model-based methods, like MPC, outperform model-free methods when an accurate world model is accessible, the practical deployment of MPC is constrained by the requirement for such a model. Crafting this world model demands additional expertise and is not straightforwardly translatable to real-world applications due to this dependency.
>    2. The selection of reward term, design reward term template, and even finding appropriate reward term weight all needs an extra large amount of expert effort.
>
> ----
>
> ### **W2: Use of open-source LLMs for reproducibility and accessibility.**
> **A:** Thanks for your suggestion! As you mentioned, it is a common practice to use the advanced GPT models from OpenAI or close source model PaLM from Google for novel applications and studies in previous work (Chain-of-Thought[1], Zero-shot planner[2], SayCan[3], ReAct[4]). For the reproducibility, we used open-source algorithms and environments and provided full reproducible prompts and code. Following your suggestion, we also add experiments in the application of our method using open-source or publicly available models on Llama-2 and Code-Llama (Llama-2 further pretrained on code corpus). We include these findings in our revised manuscript and provide a more comprehensive evaluation of our method's capabilities across different models in Appendix F. The results show that the open-source models still have a huge gap with the most advanced models in the difficult task of reward function generation and reflect the necessity of using the most advanced one to demonstrate the possibility of our method.
>
> [1] Jason Wei, Xuezhi Wang, Dale Schuurmans, et al,. Chain-of-Thought Prompting Elicits Reasoning in Large Language Models.
> https://arxiv.org/abs/2201.11903
>
> [2] Wenlong Huang, Pieter Abbeel, Deepak Pathak, Igor Mordatch. Language Models as Zero-Shot Planners: Extracting Actionable Knowledge for Embodied Agents.
> https://arxiv.org/abs/2201.07207
>
> [3] Michael Ahn, Anthony Brohan, Noah Brown, et al., Do As I Can, Not As I Say: Grounding Language in Robotic Affordances
> https://arxiv.org/abs/2204.01691
>
> [4] Shunyu Yao, Jeffrey Zhao, Dian Yu, Nan Du, Izhak Shafran, Karthik Narasimhan, Yuan Cao. ReAct: Synergizing Reasoning and Acting in Language Models.
> https://arxiv.org/abs/2210.03629
>
> ----
>
> ### **Minor Issues 1: Typographical errors and suggestions to include reward function examples in the main paper.**
> **A:** We appreciate your suggestions. The typographical errors have been corrected in our revised manuscript. While we acknowledge the value of including reward function examples for clarity, space constraints necessitate a concise presentation. Therefore, only a key example is showcased in Figure 1. We intend to include more comprehensive examples in the camera-ready version, where space permits.
>
> ----
>
> ### **Q1: Contribution compared to concurrent work, such as Eureka.**
> **A:** Eureka represents an important concurrent development in the field, we all address the important problem of dense reward specification in reinforcement learning, using a method based on zero-shot and few-shot prompting of a language model, as Reviewer#4 pointed out. We believe that our work, alongside Eureka, contributes to the broader understanding and capabilities of LLMs in policy training and reinforcement learning.

---

> > ### Comment · Reviewer_VQaY · 2023-11-16
> > **Thanks for the response**
> >
> > Thanks for the detailed response. Overall I have improved my perception of this paper after the replies and new experiments. A couple of things:
> >
> > **Comparison to L2R**
> > I like the new experiments and they resolve my concerns. But I think you should make the stated differences clearer in the related works section of the text, which hasn't been updated.
> >
> > > The use of Model Predictive Control (MPC) within the L2R framework restricts the reward term to a twice-differentiable norm with its minimum at zero, typically the L2 distance.
> >
> > Many MPC algorithms work fine without these assumptions, so I would not claim this as a differentiating factor.
> >
> > > L2R is model-based and needs additional expert effort v.s. T2R is model-free and can easily be transferred to new environments and real-world application:
> >
> > I also still do not believe this is a huge difference. L2R produces a reward term that is usable by any optimizer. That can be an MPC optimization algorithm or a traditional RL algorithm. This is simply a choice of using MPC vs RL to **optimize**, which is independent of the **quality** of the reward function, as the per-state/action rewards of T2R can be directly given to an MPC-based optimizer like CEM and vice-versa with L2R.
> >
> >
> >
> > > L2R can only use a sum of reward terms v.s. T2R can write stage and dense rewards such as if-else statements and can incorporate any other Python feature
> >
> > I now understand what the authors mean by "dense" and "sparse" rewards in this paper. But these terms are different than how they are used traditionally in RL papers, so I think the paper should be updated to correct this. Both L2R and T2R use "dense" rewards in the sense that there are non-zero, per-state rewards that are given to an optimizer. The authors are stating "dense" means the ability to use if-else conditions and other more advanced program structures, hence L2R is "sparse." But this is not how the RL community uses the word "sparse," which typically refers to not having **any** useful reward signal for most states in a trajectory. I highly suggest changing how the paper characterizes "dense" vs "sparse"; the authors should instead highlight the linear sum of reward terms of L2R vs the advanced code reward functions that T2R can produce.
> >
> > **OPen source LLMs**
> >
> > Thank you for adding this comparison, this is great.
> >
> >
> >
> > Overall I like the new version of the paper a lot better along with the many new experiments with L2R and open source LLMs. But I would still like to see some changes (or to be proven wrong in an additional author response) with respect to comparisons to prior work like L2R, as I believe some of the statements in the paper are incorrect and/or make faulty comparisons.
> >
> > I have raised my score accordingly and am willing to do so further after additional discussion.

---

> > > ### Author Response · Authors · 2023-11-17
> > >
> > > Thank you for your insightful feedback and the improved assessment of our paper. We would like to further address your concerns as follows:
> > >
> > > **Discussion on comparison to L2R in the related work**
> > >
> > > Thanks for the reminder! In the latest version, we updated the corresponding part about differences (mostly in the related work), and we also modified the “dense”/”sparse” (throughout the paper). The changed texts are shown in orange. We are happy make further clarifications throughout our discussion.
> > >
> > > **Claims on limitations of MPC (e.g., differentiable constraints, difficulty of MPC in real world)**
> > >
> > > We generally agree that the use of MPC or not should not be a differentiating factor in terms of *technical contribution*. Any reward function, dense or sparse, carefully shaped or not, can be used by either RL or MPC. Instead, we would like to say that we do believe RL, as an *experimental setting*, is the right choice to demonstrate the benefit of our LM-based, automated reward shaping. It is the minimal testbed to differentiate shaped (or, high-quality) and unshaped (or, low-quality) rewards. This is also justified by our updated experiments where unshaped rewards are challenging to train with RL on hard and multi-stage tasks.
> > >
> > > **Changing "dense" vs "sparse" to “shaped” vs “unshaped”**
> > >
> > > Your concern here is extremely valid, and we agree that “sparse reward” should be used with caution. To rectify this, we have updated the manuscript to describe our focus as “reward shaping” instead of “dense reward generation”; also from “dense”/”sparse” to “shaped”/”unshaped”. (For the paper title, we think we’ll get the chance to change it before camera ready). Relevant to our discussion, we would like to summarize our understanding of three types of rewards to check if we agree:
> > > - Sparse reward: the robot receives a binary reward only at the end of the episode, and zero otherwise.
> > > - L2R’s unshaped dense reward: the robot receives a reward at each timestep, but the working part of the reward function is the same across timesteps (but reward values are different).
> > > - Our shaped dense reward: the robot receives a reward at each timestep, and, depending on the status/stage, the reward functions are different.
> > >
> > > Please let us know if there are further points that we can clarify. Thank you for your constructive feedback!

---

> > > > ### Comment · Reviewer_VQaY · 2023-11-17
> > > > **Updated score again**
> > > >
> > > > Thanks for the response and changes.
> > > >
> > > > All of my main concerns are addressed and I have updated my score accordingly. Pending seeing the final discussion btwn the other reviewers and the authors, I currently think this paper should be accepted.

---

### Official Review · Reviewer_XwRM · 2023-10-29

**Soundness:** 3 good
**Presentation:** 3 good
**Contribution:** 3 good
**Rating:** 6
**Confidence:** 4

**Summary:**

This paper introduces Text2Reward, an automated generation framework of dense reward functions based on large language models. T2R produces interpretable, free-form dense reward codes using a compact representation of the environment source code. T2R is able to outperform human-written reward codes on a range of tasks and allow iterative refinement with human feedback.

**Strengths:**

This paper studies the pertinent problem of automated reward design using LLMs. Given that reward design is a fundamental challenge in RL and that LLMs for decision making have largely been limited to high-level planning tasks, this paper offers a fresh perspective and a nice solution to the growing literature of LLMs for problem design and low-level robotic control.

This paper's method is novel and more flexible than a prior work (Yu et al., 2023) in that it does not require extensive manual templating per robotic embodiment or task and is capable of generating free-form reward functions. It incorporates a compact representation of the environment, background knowledge (e.g., available function APIs), and/or few-shot examples to successfully do so. Finally, this paper demonstrates interesting use case of the proposed method, such as real-world evaluation as well as learning from human feedback.

The paper is well-written and free of grammatic errors.

**Weaknesses:**

1. The primary weakness of the paper is that most evaluation tasks are from benchmarks that have been released before GPT-4's knowledge cutoff date (September, 2021). Mujoco and Metaworld tasks have been extensively studied in the reinforcement learning literature; ManiSkill2, though released recently, have many overlapping tasks with ManiSkill, which was released in mid 2021; in particular, most of the tasks, to the best of my knowledge, were in the original ManiSkill benchmark. Given this, it is not clear whether the reward design capability of T2R can readily transfer to an unseen task in a new simulator.

2. Relatedly, the "novel" behavior on the Mujoco locomtoin tasks have appeared in prior literature; for example, Hopper back flip is shown in Christiano et al., 2017. It's unclear whether T2R has benefited from that knowledge possibly being in the training set of the backbone LLM.

3. Most manipulation tasks are of "pick-and-place" or opening/closing/pushing nature. These are also the most common types of manipulation tasks that the RL literature has studied. It is possible that GPT-4 is generally adept at writing reward functions for those task types.

3. T2R appears to still work best with few-shot examples. In many tasks that do not belong to a family of tasks introduced by a benchmark, providing few-shot examples can still be difficult.

3. For each task, only one reward function is reported. It is not clear whether T2R is robust to stochasticity in LLM generation.

**Questions:**

1. There are 11 tasks in MetaWorld, but only 6 of them have results in the paper?

2. Is there a reason few-shot examples are not used for MetaWorld?

3. How robust is T2R to different random generations? The temperature used for the LLM generation is not shown.

4. Can T2R work for non pick-and-place, opening/closing/pushing robotic manipulation tasks?

---

> ### Author Response · Authors · 2023-11-16
>
> Dear Reviewer XwRM,
>
> Thank you for your thoughtful review and appreciation of our work! We are glad to hear that you acknowledge our contribution to expanding the research of Large Language Models (LLMs) to low-level robotic control. Text2Reward (T2R) avoids the need for extensive manual templating for each robotic form or task and is adept at creating versatile, free-form reward functions.
>
> We have revised our manuscript and are addressing the specific questions in the following reply, including:
> - clarifications on details
> - discussion on data leakage and generality.
>
> Please let us know if you have any further questions and we can provide any additional clarifications to help finalize your assessment and rating of our paper.
>
> -----
>
> ### **W 1,2,3: Data leakage in GPT-4**
> **A:** Thanks for your question! We understand your concern regarding the potential data leakage of reward code in GPT-4 and other LLMs, which might lead one to assume that the model's ability to generate high-quality code stems from exposure to "golden examples."
>
> However, we would like to clarify three sources of evidence from our paper and previous work:
> 1. Evidence from the locomotion novel skills experiments (including backflip, frontflip, wave legs) part in our paper;
> 2. Evidence from the input side (prompt) and output side (generated reward function) of our proposed method for reward function generation;
> 3. Evidence from previous research showed the generalization capability on code generation problems of new domains that are not included in the training set.
>
> **1. Evidence from the locomotion novel skills experiments (including backflip, frontflip, wave legs) part in our paper**
>
> We did experiments on learning novel skills in MuJoCo, as shown in Table 1 and Figure 4, T2R generates reward codes under our created new instructions for novel skills. We also show the generated reward function codes in Appendix D.2.
>
> And for your concern about Christiano et al., 2017, as far as we know, although a hopper backflip was demonstrated in their work, this behavior is learned from human preferences using a neural network reward model, rather than using a symbolic reward code. Furthermore, our generated reward function (shown in Appendix D.2) is very different from what they showed as an expert-written case in their blog (https://openai.com/research/learning-from-human-preferences) of this work.
>
> **2. Evidence from the input side (prompt) and output side (generated reward function) of our proposed method for reward function generation.**
>
> **From the input end,** one piece of evidence, as shown in Appendix C, is that the prompts we provided to LLMs like GPT-4 for our T2R method are **derived from expertly abstracted class Python representations, with specific attributes and function names coined by us, not exact snippets from the ManiSkill2 or MetaWorld GitHub repositories**. This rules out the possibility of the language model plagiarizing directly through context memory since previous work [1] has attempted to perturb instructions used for code generation and found significant performance drops.
>
> On the other hand, we also have preliminary results that entering direct commands like "generate a reward function for crawler" or "pick the cube" into GPT-4, or inputting the ManiSkill2 environment code context into GPT-4 to let it complete the `compute_dense_reward` section, none of which were successful. This not only demonstrates the inadequacy of mere memorization but also validates the effectiveness of our method.

---

> > ### Author Response · Authors · 2023-11-16
> >
> > **From the output end,** another piece of evidence is shown in Appendix D.1 of our paper, where we compare instances of code generated by LLM through zero-shot and few-shot prompting with the oracle code example on the ManiSkill2 Pick Cube task.
> >
> > Below is the code generated by zero-shot prompting GPT-4 using the T2R method:
> > ```python
> > import numpy as np
> >
> > def compute_dense_reward(self, action):
> >     # Normalize action
> >     action = np.clip(action, -1, 1)
> >
> >     # Calculate distance between gripper and cube
> >     gripper_pos = self.robot.ee_pose.p
> >     cube_pos = self.cubeA.pose.p
> >     dist_gripper_cube = np.linalg.norm(gripper_pos - cube_pos)
> >
> >     # Calculate distance between cube and goal
> >     goal_pos = self.goal_position dist_cube_goal = np.linalg.norm(goal_pos - cube_pos)
> >
> >     # Check if the robot is grasping the cube
> >     grasping_cube = self.robot.check_grasp(self.cubeA)
> >
> >     # Define reward components
> >     reward_dist_gripper_cube = -1.0 * dist_gripper_cube
> >     reward_dist_cube_goal = -1.0 * dist_cube_goal
> >     reward_grasping_cube = 1.0 if grasping_cube else -1.0
> >
> >     # Define weights for reward components
> >     weight_dist_gripper_cube = 0.3
> >     weight_dist_cube_goal = 0.5
> >     weight_grasping_cube = 0.2
> >
> >     # Calculate total reward
> >     reward = weight_dist_gripper_cube * reward_dist_gripper_cube \
> >                 + weight_dist_cube_goal * reward_dist_cube_goal \
> >                 + weight_grasping_cube * reward_grasping_cube
> >
> >     # Regularization on action
> >     reward -= 0.01 * (action ** 2).sum()
> >     return reward
> > ```
> >
> > And here is the oracle code (from the experts of the ManiSkill team):
> > ```
> > import numpy as np
> >
> > def compute_dense_reward(self, action):
> >     reward = 0.0
> >
> >     is_obj_placed = np.linalg.norm(self.goal_position - self.cubeA.pose.p) <= 0.025
> >     is_robot_static = np.max(np.abs(self.robot.qvel)) <= 0.2
> >
> >     success = is_obj_placed and is_robot_static
> >
> >     if success:
> >         reward += 5
> >         return reward
> >
> >     tcp_to_obj_pos = self.cubeA.pose.p - self.robot.ee_pose.p
> >     tcp_to_obj_dist = np.linalg.norm(tcp_to_obj_pos)
> >     reaching_reward = 1 - np.tanh(5 * tcp_to_obj_dist)
> >     reward += reaching_reward
> >
> >     is_grasped = self.robot.check_grasp(self.cubeA, max_angle=30)
> >     reward += 1 if is_grasped else 0.0
> >
> >     if is_grasped:
> >         obj_to_goal_dist = np.linalg.norm(self.goal_position - self.cubeA.pose.p)
> >         place_reward = 1 - np.tanh(5 * obj_to_goal_dist)
> >         reward += place_reward
> >
> >     return reward
> > ```
> >
> > If the process were reliant on memory replication, these two code segments would be very similar. However, here the styles and approaches of the reward are quite different, yet they ultimately achieve the same effect. Thus, we wouldn’t say the "leakage" argument negates the efficacy of the T2R method.
> >
> > **3. Evidence from previous research showed the generalization capability on code generation problems of new domains that are not included in the training set**
> >
> > Previous research has shown the generalization capability and great performance on code generation problems that are not included in the training set of GPT-4 by conducting:
> >
> >  - contamination study [2]: According to Appendix C (Contamination on professional and academic exams), Appendix D (Contamination on academic benchmarks), Table 9, Table 10, and Table 11, the GPT-4 team from OpenAI trained different versions of GPT-4 w. and w.o. the content from the evaluation set and conclude “contamination overall has very little effect on the reported results.”.
> >  - testing on fresh coding questions after GPT-4’s pretraining period [3]. They construct a benchmark of 100 LeetCode problems posted after October 8th, 2022, which is after GPT-4’s pretraining period and show that GPT-4 significantly outperforms the other models and is comparable to human performance on these fresh questions.
> >
> > [1] Yuhang Lai, Chengxi Li, Yiming Wang, Tianyi Zhang, Ruiqi Zhong, Luke Zettlemoyer, Scott Wen-tau Yih, Daniel Fried, Sida Wang, Tao Yu. DS-1000: A Natural and Reliable Benchmark for Data Science Code Generation https://arxiv.org/abs/2211.11501
> >
> > [2] OpenAI, GPT-4 Technical Report
> > https://arxiv.org/abs/2303.08774
> >
> > [3] Sébastien Bubeck, Varun Chandrasekaran, Ronen Eldan, Johannes Gehrke, Eric Horvitz, Ece Kamar, Peter Lee, Yin Tat Lee, Yuanzhi Li, Scott Lundberg, Harsha Nori, Hamid Palangi, Marco Tulio Ribeiro, Yi Zhang. Sparks of Artificial General Intelligence: Early experiments with GPT-4
> > https://arxiv.org/abs/2303.12712

---

> > > ### Author Response · Authors · 2023-11-16
> > >
> > > ### **W4: Consideration of the difficulty when using few-shot T2R in new domain**
> > > **A:** Thank you for the inquiry. We want to clarify that few-shot examples, while beneficial, are not a prerequisite for the success of our T2R approach, given the current strength of LLMs. Our MetaWorld zero-shot setting experiments demonstrate that the rewards generated are on par with those crafted by human experts and are effective for developing novel locomotion skills. We included few-shot setting experiments as a means to enhance performance, aligning with established practices in in-context learning and code generation research [1, 2, 3].
> > >
> > > [1] Tom B. Brown, Benjamin Mann, Nick Ryder, et al., Language Models are Few-Shot Learners
> > > https://arxiv.org/abs/2005.14165
> > >
> > > [2] Takeshi Kojima, Shixiang Shane Gu, Machel Reid, Yutaka Matsuo, Yusuke Iwasawa, Large Language Models are Zero-Shot Reasoners
> > > https://arxiv.org/abs/2205.11916
> > >
> > > [3] Mark Chen, Jerry Tworek, Heewoo Jun, et al., Evaluating Large Language Models Trained on Code
> > > https://arxiv.org/abs/2107.03374
> > >
> > > -------
> > >
> > > ### **W5: Not clear whether T2R is robust to stochasticity in LLM generation**
> > > **A:** Thank you for raising this point. In our error analysis presented in Table 2, we examined 100 reward function examples generated by both zero-shot and few-shot prompts across 10 different Maniskill2 tasks. This involved a detailed review of 10 unique reward codes generated by a language model per task with the same prompt to assess accuracy. While policy training for success rate stability could offer deeper robustness insights, our current aim is to showcase T2R's capability to simplify policy learning. We'll consider adding robustness experiments in the camera-ready version.
> > >
> > > ------
> > >
> > > ### **Q1: Why are only 6 out of the 11 tasks in MetaWorld reported in the paper?**
> > > **A:**  To clarify, we include 10 tasks from Metaworld in the same fashion as follows [1] and [2]. Among all 17 included tasks, 7 are from Maniskill and 10 (instead of 11) are from Metaworld. The 7 figures for Maniskill tasks are 6 subfigures in Figure 2 and 1 figure in Figure 7 (to avoid replotting the same result twice). The 10 figures for Metaworld are 6 subfigures in Figure 3 and 4 subfigures in Figure 8 in the Appendix (to save space in the main text).
> > >
> > > [1] Kimin Lee, Laura Smith, Pieter Abbeel. PEBBLE: Feedback-Efficient Interactive Reinforcement Learning via Relabeling Experience and Unsupervised Pre-training https://arxiv.org/abs/2106.05091
> > >
> > > [2] Joey Hejna, Dorsa Sadigh. Few-Shot Preference Learning for Human-in-the-Loop RL https://arxiv.org/abs/2212.03363
> > >
> > > -------
> > >
> > > ### **Q2: Is there a specific reason why few-shot examples were not used for MetaWorld?**
> > > **A:** In MetaWorld, given its relatively straightforward task set with simple reward function expressions, we found that most challenges could be addressed through zero-shot prompting. This underscores the potential for generating effective rewards by relying solely on a compact representation of the environment.
> > >
> > > -------
> > >
> > > ### **Q3: How robust is T2R to different random generations? What temperature setting was used for LLM generation?**
> > > **A:** Thank you for your question! In line with what we discussed in response to W5, we've indeed touched upon this in Table 2. We'd like to point out that while LLM generation robustness is interesting, our primary goal here is to illustrate how T2R can facilitate the simplification of policy learning.
> > > For GPT-4, we set the temperature to 0.7 for all the experiments.
> > >
> > > ------
> > >
> > > ### **Q4: Can T2R work for non pick-and-place, opening/closing/pushing robotic manipulation tasks?**
> > > **A:** In theory, our formulation can do any task if its reward can be written in the format of code. Technically, for 2-finger parallel gripper manipulation, we believe our method can successfully work for assembly tasks such as peg-in-the-hole. To apply our method to dexterous hand manipulation, as well as deformable object (soft-body, cloth, etc.) manipulation, one can extend our reward to compute the distance between meshes and fingers.

---

> > > > ### Comment · Reviewer_XwRM · 2023-11-22
> > > >
> > > > Dear Authors,
> > > >
> > > > I have read your responses and believe that most of my concerns have been adequately addressed. That said, I think the approach is still limited in the sense that the algorithm does not systematically handle the case when the reward generation is not good enough either with zero-shot or few-shot prompting. Providing few-shot examples for reward functions also appear more difficult than other use cases of few-shot prompting. For these reasons, given my initial high score, I have elected to keep my rating.

---

### Official Review · Reviewer_fmuH · 2023-10-31

**Soundness:** 2 fair
**Presentation:** 2 fair
**Contribution:** 3 good
**Rating:** 8
**Confidence:** 3

**Summary:**

The paper proposes a method to generate dense rewards for continuous control RL tasks using LLMs and an API summary of the environment. The method runs either completely zero-shot on top of the prompt with environment information, or in a few-shot manner by concatenating the nearest neighbor task specification and expert reward into the context. In addition, after RL training, human feedback can be solicited, leading to further refinement of the reward code by the LLM, which leads to hopefully stronger RL. Across a number of tasks, the zero-shot dense reward code is competitive with the expert code, and in some settings few-shot generation can improve the results, though in others it is harmful.

**Strengths:**

- Interesting application of a new tool! Using LLMs to generate reward code seems like an easy way to simplify problems we may not already have solutions for, but can describe in language, and is a completely different way around the sparse reward problem
- Zero-shot results seem pretty strong across all environments
- Nice results on new tasks that (as far as I know) we don't have expert reward for (e.g. Hopper flipping)

**Weaknesses:**

- There is no qualitative analysis/discussion of what the source of the improvement is:
  - Why does Zero-shot outperform Few-shot on Turn Faucet, Open Cabinet Door, Open Cabinet Drawer?
  - Why does Few-shot fail to outperform Oracle even though Oracle is in context (Lift Cube, Turn Faucet, Push Chair, Open Cabinet Drawer)?
  - In the cases that few-shot improves on zero-shot, what is the source of this improvement?
- There are a lot of missing details in the experiments:
  - There is only one example of human feedback (Figure 6) and it is in a schematic diagram for a task that does not have ground truth, I would like to see a few traces of the whole round (generation, feedback, generation, feedback, generation) in order to understand what exactly is happening
  - For results in Table 1 and Figure 4 on novel behaviors, the standards for human evaluation and who conducted the evaluation (the authors, peers from the same institution, crowdsourced to a platform) are missing
  - The experiments in Table 1 and Figure 4 are supposedly conducted in a zero-shot setting (caption in Figure 4), yet Figure 6 gives a schematic for ambiguity resolution which would imply a few-shot result for novel behaviors, an experiment which I do not see in the paper and appendices.
  - I do not see the choice of $k$ for the number of neighbors that appear in the few-shot context
  - For generated rewards in Appendix D.1 on Pick Cube, the few-shot vs. oracle code is almost indistinguishable except for 2 constants (cube placed at goal reward, grasped reward). Given this difference is so small, it seems important to know what the human feedback was: are we just getting lucky?
  - As before, given that the few-shot vs. oracle code is so close on Pick Cube (the only example we have to judge), why is it the case that the few-shot generation is underperforming oracle generation in other settings (Lift Cube, Turn Faucet, Open Cabinet Drawer, Push Chair)?
  - Is it always the case that the Oracle code for the task is put into the context for Few-shot?
- Section 4.2 is about code errors that occur before any RL happens, and this seems like a necessary filtering step, but I think having an example of the generation/feedback process is much more important in the main body than Table 2.
- How long does the iteration loop take? Each iteration requires training policies, so it is quite expensive, and it may be nice to think about early evaluation
- It would be nice to include code examples for the novel behavior tasks to see what is happening. Given there is no baseline in this case, simply presenting a quantitative evaluation without any analysis is a little sparse...

**Questions:**

See weaknesses above, I'm most interested in analysis on the source of the improvements in each of the tasks, and how the language model goes about creating reward for a novel task

---

> ### Author Response · Authors · 2023-11-16
>
> Dear Reviewer fmuH,
>
> Thank you for your careful review and constructive comments. We are glad you describe our paper as an interesting application with nice results across environments.
>
> At a very high level, we would like to clarify several terms to make sure that we are on the same page. By human oracle we mean the human-written dense rewards provided by the environment (tuned by the authors of the environment paper). By few-shot, we mean using related human-written dense rewards as in-context reward examples, so few-shot itself does not include any human feedback or interactive generation. Furthermore, in the few-shot setting, for each task, we remove its oracle code from the retrieval pool to make sure that the LM does not cheat.
>
> We have revised our manuscript and are addressing the specific questions in the following reply, including:
>  - Clarification on details
>  - Additional examples provided in the paper
>
> Please let us know if you have any further questions and we can provide any additional clarifications to help finalize your assessment and rating of our paper.
>
> ----
>
> ### **W1: There is no qualitative analysis/discussion of what the source of the improvement is:**
>
> **W1.1 & W1.2: Why does zero-shot sometimes outperform few-shot? Why does few-shot fail to outperform oracle even though oracle is in context?**
>
> **A:** Thanks for your question! We observed that the LM tends to copy code snippets from the few-shot code when provided. If the human oracle is not optimal, the few-shot human oracles might inadvertently introduce biases or constraints, limiting the LM's creative reward generation.
>
> **W1.3: In the cases that few-shot improves on zero-shot, what is the source of this improvement?**
>
> **A:** Robotic tasks are related, especially the dense stages rewards. When few-shot improves on zero-shot, the few-shot examples likely provide relevant, task-specific stage rewards based on which the LM can rewrite.
>
> ------
>
> ### **W2: There are missing details in the experiments:**
>
> **W2.1: Clarification on the human feedback process and its trace in the experiments.**
>
> **A:** In the revised manuscript Appendix D.3, we added a detailed example showing the traces of the whole round to show how human feedback improves the reward code step by step.
>
> **W2.2: Human evaluation for locomotion**
>
> **A:** The authors evaluated each rollout. We add this detail to the revised manuscript.
>
> **W2.3: Clarification on the setting of Figure 6**
>
> **A:** The locomotion example in Figure 6 is zero-shot + human feedback, but not few-shot (because we do not have a human oracle for all locomotion tasks).
>
> **W2.4: The choice of $k$ for the number of neighbors in few-shot context.**
>
> **A:** We set $k$ as 1 in all experiments of few-shot settings.
>
> **W2.5 & W2.6 & W2.7: Few-shot and oracle are sometimes non-distinguishable. Few-shot sometimes underperform oracle. Is it always the case that the Oracle code for the task is put into the context for Few-shot?**
>
> **A:** We design the experiment so that the oracle code for the task is never in the context or few-shot. For each task, we remove its oracle code from the retrieval pool to make sure that the LM does not cheat. This explains why few-shot is not necessarily better than oracle because it cannot copy the oracle. When few-shot and oracle are similar, one potential reason is that there is a similar task in the few-shot example, and the LM can adapt that oracle code to fit into this task.
>
> -------
>
> ### **W3: The importance of code errors section vs. generation/feedback process details.**
>
> **A:** Thank you for your suggestion! We wish to clarify that when using GPT-4, syntax errors are generally not encountered, as indicated in Table 2, where direct generation has only a 10% error rate. Furthermore, we found that by feeding the error messages from the cases with syntax errors back to GPT-4, all such errors can be rectified. Given that it has become commonplace to use error messages to guide GPT-4 in correcting syntax issues, especially considering that reducing syntax error rates is not the focus of our paper, we choose to not add this as a section.
>
> ------
>
> ### **W4: Duration of the iteration loop and consideration of early evaluation.**
>
> **A:** We provide information on the time taken for each iteration loop with the hardware we used and discuss the potential for early evaluation strategies to improve efficiency in Appendix A.
>
> ------
>
> ### **W5: Inclusion of code examples for novel behavior tasks.**
>
> **A:** Just as you mentioned, to enhance clarity and understanding, we include reward code examples generated for novel behavior tasks in Appendix D.2. This will provide a clearer picture of how the LM approaches these tasks and the nature of the generated rewards.

---

> > ### Comment · Reviewer_fmuH · 2023-11-20
> > **Thanks for your response**
> >
> > Thanks for your response, many of my questions have been addressed in a limited manner. I still think there's relatively little clarity as to what is actually happening:
> >
> > > There is no qualitative analysis...
> >
> > The included results are in the appendix without accompanying discussion. At the very least, something of this nature should be in the main text because just numbers hide any ability to build on top of your work. This goes for both subpoints.
> >
> > > There are missing details...
> >
> > Thanks for some of these changes. The choice of k should be reflected in the section and Table titles in the main text given it's consistent always. In addition the details on the fact that the oracle code is never in context seems like it should be in the main text, and it makes the close agreement between the few-shot and oracle examples even more confusing: it seems that the oracle code is getting leaked in this scenario.
> >
> > > The importance of code errors sections vs. generation/feedback process details.
> >
> > I agree that checking syntax is not an important contribution of the work, which is why I advocated for including something like what is now Appendix D.3 in the main text. I generally believe that GPT can correct its own execution errors, and this doesn't seem fundamental to your work, so the choice of Section 4.2 is odd to me.
> >
> > > Duration of the iteration loop...
> >
> > Thanks for these clarifications.
> >
> > > Inclusion of code examples...
> >
> > Thanks for the inclusions. I should note that I've become aware after the fact that it's possible to make hopper front and backflip with only a constant policy, and the same is true of the ant lying down, so I'm not sure how convincing these are as novel tasks any longer, but I still consider the paper interesting.

---

> > > ### Author Response · Authors · 2023-11-21
> > >
> > > Thank you for your kind feedback and constructive comments. We have revised the manuscript (updates to your points are in blue color) and addressed your concerns as follows:
> > >
> > > **There is no qualitative analysis**
> > >
> > > For your concerns regarding the qualitative analysis, we have moved the error analysis section (which we now agree is of relative unimportance) to the Appendix and put a new qualitative analysis section there. Space limited, we refer the readers to Appendix D for the complete reward functions, and we discuss how zero-shot, few-shot, and oracle rewards are different and why in the main text. This offers insights into questions such as "what may be the source that makes few-shot setting generation better", "what causes the zero-shot setting to sometimes generate better outcomes than the few-shot one", and "why sometimes even the expert-written oracle reward function is not the best".
> > >
> > > **Detail clarification**
> > >
> > > For the few-shot examples, we have revised the manuscript to include clear classifications not only in the Appendix but also in Section 2.2, and in the captions of Figures 2 and 7.
> > >
> > > **The importance of code errors sections vs. generation/feedback process details**
> > >
> > > As mentioned, we have moved this part into the Appendix and conducted a more informative qualitative analysis in the main body. Thank you for your suggestions!
> > >
> > > Thank you for your constructive feedback! Feel free to put any other thoughts before the discussion closes.

---

> ### Comment · Reviewer_fmuH · 2023-11-22
>
> > For your concerns regarding the qualitative analysis...
>
> I much prefer this change. I do think that this section should refer to both precise performance curves in Figure 2 and examples of generated code in the Appendix (where included, like with Pick Cube) as this makes the information location job much easier on the reader. I don't think this section answers my previous concern that oracle code was being leaked in the case of Pick Cube, and I still don't have a response to this...
>
> > Detail clarification
>
> I appreciate these changes.
>
> > The importance of code errors...
>
> I think the structure is more informative now.
>
> I think given the many changes I would currently support changing my score from 6 to 8. I would prefer seeing the minor change above before the camera ready, though I realize discussion closes soon.

---

> > ### Author Response · Authors · 2023-11-23
> >
> > Dear Reviewer fmuH,
> >
> > Thanks again for your detailed and insightful comments! Due to current time constraints, we are unable to update these minor change immediately, but we assure you that we will definitely address all suggested changes diligently for the camera-ready version. Your understanding is greatly appreciated!

---

### Author Response · Authors · 2023-11-16
**Updated Manuscript and Response to All Reviewers**

We sincerely thank all the reviewers for their feedback and constructive comments. We are pleased that the reviewers appreciate finding our method shows a completely different way something not shown around the sparse reward problem (R#1, R#4), provides a fresh perspective and a nice solution to the growing literature of LLMs for problem design and low-level robotic control (R#2), nice/solid results across comprehensive tasks including real-world applications (R#1, R#2, R#3, R#4), free of grammatical errors in paper writing (R#2).

We’ve made an **additional 3 experiments (2 baselines and 2 new open-source models) and revised and updated the manuscript**, which reflects the reviewers' comments (all revisions are highlighted in **purple color** in the new PDF). The updates are summarized as follows:

1. **Section 3.1:** Add a description of the additional experiments on L2R to further demonstrate the effectiveness of our method (R#3).

2. **Section 4.1:** Include the experimental results of our newly added L2R baselines on 6 tasks of ManiSkil2, and provide a comparison (R#3).

3. **Table 1:** Insert details of the human evaluation process into the table caption (R#1).

4. **Appendix A:** Provide more details on the duration of the iteration loop and the consideration of early evaluation (R#1), and include details of the temperature setting used for LLM generation (R#2).

5. **Appendix D.2:** Add examples of reward code functions to include code samples for novel behavior tasks (R#1).

6. **Appendix D.3:** Enhance the section with additional details for clarification on the human feedback process and its role in the experiments (R#1).

7. **Appendix E:** Conduct further experiments on L2R on 6 tasks of ManiSkill2 and provide analysis to demonstrate the effectiveness of our approach (R#3).

8. **Section 3 & Appendix F:** Present further experiments of applying our methods on open-sourced language models Llama-2 and Code-Llama on 10 tasks of MetaWorld, and display the results (R#3, R#4).

We have addressed the queries and points raised by each reviewer below and made replies for clarification. Our sincere thanks go out once again to all reviewers for their valuable contributions towards enhancing our manuscript. Should there be a need for further clarification to assist in advancing our score, please do not hesitate to inform us.

Thank you for your review!

---

### Meta-Review · Area_Chair_149z · 2023-12-05

**Metareview:**

**Summary**: The paper proposes T2R, a method to generate dense rewards using LLMs, given a textual description of the environment. The method makes use of expert provided python class declarations combined with human-provided text instructions of the desired task, to prompt GPT-4 into generating python code for dense rewards. This prompt can optionally be augmented with few-shot examples of expert and previously generated reward functions that are "close" based on an embedding similarity between desired and stored instructions. The framework also allows for iterative improvement of the reward by having a human provide feedback given a video from a policy learned with the current reward. The method is evaluated across a number of tasks and a variety of simulated environments, and the results provide evidence that the generated rewards are competitive with expert code.

**Strengths**:
- The idea of applying LLMs to generate dense reward code is interesting. Prior work has seen success in LLM-generated sparse rewards, but dense rewards are much more useful and complex to pull off.
- The method is evaluated comprehensively: 17 tasks in ManiSkill2 and Metaworld, 6 in Mujoco, and also a few real world experiments.
- The zero-shot results are strong across all environments, showcasing that the method is competitive with human expert reward specification. The results are also compelling on tasks that we can't easily design expert rewards for (e.g. Hopper flipping).
- The paper is well-written and free of grammatical errors.
- The authors also include an interesting analysis of the failure modes of reward specification, specifically when the generated reward function code leads to a python runtime error.

**Weaknesses**:
- It's unclear if T2R can transfer to more complex tasks, since most manipulation tasks are "pick-and-place" or opening/closing/pushing.
- No truly new environments are tackled in this paper. Since GPT-4 can be reasonably expected to draw from training data that already has numerous reward function for these known environments, it's unclear how well T2R would transfer to new environments.
- T2R still works best with few-shot examples, but these few-shot examples can be really difficult or impossible to provide.
- The reliance on GPT-4 makes the results inherently irreproducible.

Overall, despite these weaknesses, I agree with the reviewers that this work is comprehensively evaluated and exhibits impressive expert-level performance that merits publication.

**Justification For Why Not Higher Score:**

There still are some notable weaknesses: 1. How well does this method transfer to more complex tasks? 2. Can this method transfer to new environments -- else how do we know that the model didn't just memorize from the plethora of reward functions for these extremely well studied environments? 3. Since few-shot still works better, how do we make sure designers can even explicate these examples in code?

**Justification For Why Not Lower Score:**

The method is comprehensively evaluated and exhibits impressive expert-level performance that merits a spotlight, in my opinion.

---

### Decision · Program_Chairs · 2024-01-16

Accept (spotlight)